# Ubiquitin ligase activity inhibits Cdk5 to control axon termination

**Muriel Desbois**[1], **Karla J. Opperman**[1], **Jonathan Amezquita**[1,2], **Gabriel Gaglio**[3], **Oliver Crawley**[4], **Brock Grill**[1,5,6]*

1 Center for Integrative Brain Research, Seattle Children's Research Institute, Seattle, Washington, United States of America, 2 Molecular and Cellular Biology Graduate Program, University of Washington, Seattle, Washington, United States of America, 3 Department of Neuroscience, The Scripps Research Institute, Jupiter, Florida, United States of America, 4 Instituto de Neurociencias de Alicante, Universidad Miguel Hernández-CSIC, San Juan de Alicante, Spain, 5 Department of Pediatrics, University of Washington School of Medicine, Seattle, Washington, United States of America, 6 Department of Pharmacology, University of Washington M1-A303/305 Behnke Conference Room, Arnold building, Seattle, Washington, United States of America

* brock.grill@seattlechildrens.org

**Data Availability Statement:** All relevant data are within the manuscript and its Supporting Information files.

**Funding:** B.G. was supported by the National Institutes of Health Grant R01 NS072129. O.C. was

## Abstract

The Cdk5 kinase plays prominent roles in nervous system development, plasticity, behavior and disease. It also has important, non-neuronal functions in cancer, the immune system and insulin secretion. At present, we do not fully understand negative regulatory mechanisms that restrict Cdk5. Here, we use *Caenorhabditis elegans* to show that CDK-5 is inhibited by the RPM-1/FSN-1 ubiquitin ligase complex. This atypical RING ubiquitin ligase is conserved from *C. elegans* through mammals. Our finding originated from unbiased, *in vivo* affinity purification proteomics, which identified CDK-5 as a putative RPM-1 substrate. CRISPR-based, native biochemistry showed that CDK-5 interacts with the RPM-1/FSN-1 ubiquitin ligase complex. A CRISPR engineered RPM-1 substrate 'trap' enriched CDK-5 binding, which was mediated by the FSN-1 substrate recognition module. To test the functional genetic relationship between the RPM-1/FSN-1 ubiquitin ligase complex and CDK-5, we evaluated axon termination in mechanosensory neurons and motor neurons. Our results indicate that RPM-1/FSN-1 ubiquitin ligase activity restricts CDK-5 to control axon termination. Collectively, these proteomic, biochemical and genetic results increase our understanding of mechanisms that restrain Cdk5 in the nervous system.

## Author summary

Cdk5 is an atypical cyclin dependent kinase and an important player in nervous system development, plasticity, and disease. Decades of research has focused on understanding how Cdk5 is activated. In contrast, we know much less about the genetic and molecular mechanisms that restrict Cdk5 activity. Here, we examined how Cdk5 is inhibited in the nervous system using the model organism *C. elegans*. Our results indicate that the RPM-1/FSN-1 E3 ubiquitin ligase complex inhibits Cdk5 to control termination of axon growth. Our finding that ubiquitin ligase activity restricts Cdk5 in the nervous system *in vivo* now

supported by the Severo Ochoa Postdoctoral Program and a Marie Sklodowska-Curie Actions Individual Fellowship. The funders had no role in study design, data collection and analysis, decision to publish, or preparation of the manuscript.

**Competing interests:** The authors have declared that no competing interests exist.

opens up the interesting possibility that ubiquitin ligase activity might regulate Cdk5 in other cellular contexts and disease settings.

## Introduction

Cdk5 is a highly conserved, atypical cyclin-dependent kinase that has broad roles in the nervous system. This includes functions in multiple steps of neuronal development (neuron migration, neurite outgrowth and synapse formation), as well as effects on synaptic plasticity and behavior [1]. Abnormal Cdk5 activity is implicated in several neurodegenerative diseases including Alzheimer's disease, Parkinson's disease and Huntington's disease [2,3]. Growing evidence indicates that Cdk5 has important non-neuronal functions in cancer progression, insulin secretion, and immune responses [4,5]. These prominent, broad Cdk5 functions highlight the importance of understanding how Cdk5 is regulated.

More than two decades of research has revealed how Cdk5 is activated [6,7]. This primarily occurs via p35 and p39, which bind to Cdk5 and form activated kinase complexes [8–10]. Proteolytic processing of p35 by Calpain generates p25, a Cdk5 activator involved in neurodegeneration [11,12]. Several other activating mechanisms for Cdk5 have been identified. Examples include the tyrosine kinases Abl and Fyn [13–15], Eph A4 signaling [16], and cyclin I [17]. Post-translational modification by S-nitrosylation can activate Cdk5, but effects of S-nitrosylation can vary depending upon which Cdk5 residue is modified [18–20].

Previous studies identified three negative modulators of Cdk5: cyclin E, cyclin D1, and GSTP1 [21–23]. These molecules all share a similar inhibitory mechanism whereby they compete with p35 and p39 for binding to Cdk5, and form inactive Cdk5 complexes. Cyclin E is the most extensively studied in the nervous system, where it affects synapse density and memory formation [21]. At present, we know very little about other molecular players and mechanisms that inhibit Cdk5.

The ubiquitin-proteasome system plays an important role inhibiting protein function and signaling. Specificity of the ubiquitin-proteasome system is generated by E3 ubiquitin ligases, enzymes that recognize and transfer ubiquitin to substrates [24,25]. In the nervous system, ubiquitin ligases are key regulators of neuron development and function, and extensively involved in both neurodegenerative diseases and neurodevelopmental disorders [26–28]. At present, our knowledge about the relationship between ubiquitin ligases and Cdk5 remains limited. Cell-based studies showed that Cdk5 is ubiquitinated, ubiquitination increases with cell cycle re-entry, and overexpression of the Chd1 ubiquitin ligase affects Cdk5 levels via the proteasome [29]. Despite this valuable progress, several key questions remain unanswered in any system. Do ubiquitin ligases interact with and regulate Cdk5 *in vivo*? Does ubiquitin ligase activity regulate Cdk5 in the nervous system?

One ubiquitin ligase with important, conserved roles in nervous system development and degeneration is RPM-1, an atypical RING E3 ubiquitin ligase [30–33]. RPM-1 forms a multi-subunit ubiquitin ligase complex with the F-box protein FSN-1, which is part of the substrate recognition module [34–36]. The RPM-1/FSN-1 ubiquitin ligase complex in *Caenorhabditis elegans* is orthologous to the mammalian ubiquitin ligase complex formed by the human RING protein MYCBP2 (mouse Phr1) and the FBXO45 F-box protein. Using *in vivo* proteomics in *C. elegans* and a biochemical substrate 'trap', we identified CDK-5 as a putative RPM-1 substrate. CRISPR/Cas9-based, native biochemistry in combination with CRISPR editing confirmed that CDK-5 displays substrate-like interactions with the RPM-1/FSN-1 ubiquitin ligase complex. The functional genetic relationship between the RPM-1/FSN-1 ubiquitin ligase complex and CDK-5 was evaluated using traditional genetics, transgenics and CRISPR/Cas9

editing. Our results indicate that RPM-1/FSN-1 ubiquitin ligase activity inhibits CDK-5 to regulate axon termination in mechanosensory neurons and motor neurons. These findings provide important new insight into the biochemical and genetic mechanisms that restrict Cdk5 activity in the nervous system.

## Results

### Proteomics identifies CDK-5 as putative substrate for the RPM-1 ubiquitin ligase

To identify new substrates for the RPM-1 ubiquitin ligase, we previously deployed unbiased, large-scale affinity-purification proteomics using *C. elegans* [33]. To enrich putative substrates, we relied upon an RPM-1 ubiquitin ligase dead (LD) construct that is catalytically inactive (**Fig 1A**). We previously demonstrated that the RPM-1 LD acts as a biochemical 'trap' that enriches substrates without ubiquitinating them or triggering subsequent proteasome-mediated degradation (**Fig 1A**) [33]. There are several other notable details about our approach. Proteomics was performed with RPM-1 LD or a wild-type RPM-1 construct fused to a protein G::Streptavidin binding peptide (GS) affinity tag, which was expressed as an integrated transgene on an *rpm-1* null background. Putative RPM-1 substrates were enriched in GS::RPM-1 LD compared to GS::RPM-1 samples. Finally, transgenic GS::GFP was used as a negative control and expressed similarly to RPM-1 constructs.

Here, we analyzed this large-scale proteomic dataset and identified another putative RPM-1 substrate, the serine/threonine kinase CDK-5 (**Fig 1**). In experiments containing CDK-5, we also confirmed enrichment of UNC-51(ULK1) with GS::RPM-1 LD compared to GS::RPM-1 samples (**Fig 1B**). This is consistent with our prior study that biochemically and functionally validated UNC-51 as an RPM-1 substrate that is ubiquitinated and degraded by the proteasome [33]. Also shown in Fig 1B are several known RPM-1 binding proteins present in both RPM-1 LD and RPM-1 samples. We detected the FSN-1 F-box and SKR-1(SKP1), which form the substrate recognition module of the atypical RING ubiquitin ligase complex formed by RPM-1 (**Fig 1B**) [33,34,36]. We further identified GLO-4, RAE-1 and PPM-2, which are not substrates and mediate RPM-1 signaling (**Fig 1B**) [37–39].

CDK-5 was identified in GS::RPM-1 LD samples (**Fig 1B**), but not in GS::RPM-1 samples (**Fig 1B**) or GS::GFP negative control samples (**Fig 1C**). Analysis of cumulative results from 7 independent proteomic experiments showed that CDK-5 was significantly and exclusively identified in GS::RPM-1 LD samples from 3 experiments compared to both GS::RPM-1 or GS::GFP (**Fig 1E**). We identified four unique CDK-5 peptides that yielded 13% sequence coverage (**Figs 1D, 1E, S1A and S1B**). The CDK-5 activator CDKA-1 (homologous to mammalian p35 and p39) was also identified in GS::RPM-1 LD samples from 2 experiments (**Figs 1B, 1C, S1C and S1D**). However, CDKA-1 was detected at lower levels and was not significant compared to GS::RPM-1 or GS::GFP (**Fig 1E**).

Thus, *in vivo* affinity purification proteomics using *C. elegans* demonstrated that CDK-5 is exclusively present in GS::RPM-1 LD substrate 'trap' samples compared to GS::RPM-1 or negative control samples. This indicates that RPM-1 interacts with CDK-5 and suggests that CDK-5 is a potential RPM-1 substrate.

### CRISPR-based, native biochemistry validates CDK-5 substrate-like interactions with the RPM-1/FSN-1 ubiquitin ligase complex

To further test whether CDK-5 is a possible substrate for the RPM-1/FSN-1 ubiquitin ligase complex, we turned to native biochemistry using CRISPR/Cas9 engineered *C. elegans*.

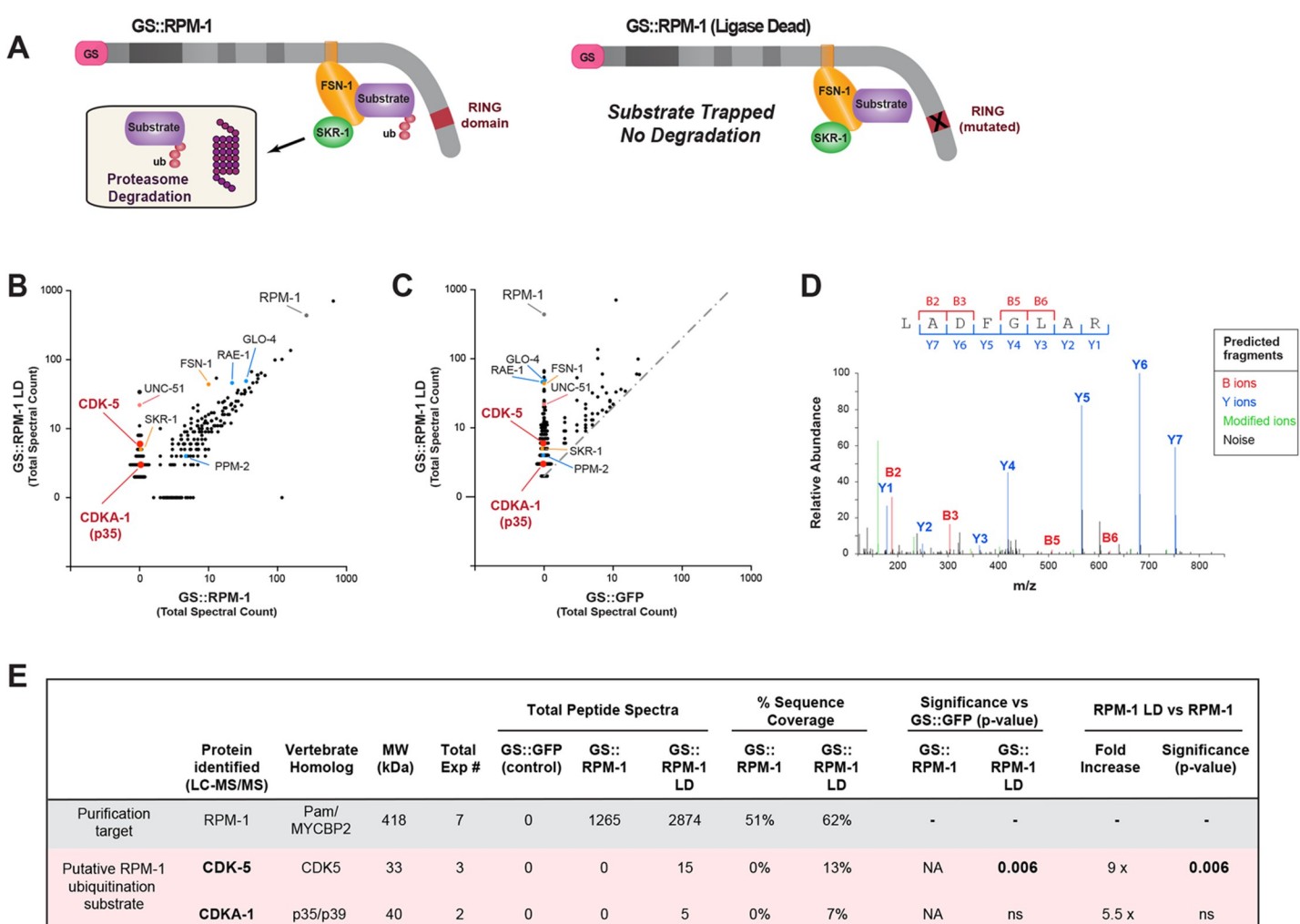

**Fig 1. Proteomics from *C. elegans* identifies CDK-5 as a putative substrate for the RPM-1 ubiquitin ligase. A)** Schematic of RPM-1 ubiquitin ligase complex and GS tagged constructs used for affinity purification proteomics. GS::RPM-1 LD substrate 'trap' does not ubiquitinate substrates which can still bind to the FSN-1 substrate recognition module. As a result, RPM-1/FSN-1 ubiquitination substrates are enriched, but not ubiquitinated and not degraded by the proteasome. Note transgenic GS:: RPM-1 LD is generated on an *rpm-1* null background to facilitate substrate enrichment and interactions with the endogenous, untagged ubiquitin ligase components FSN-1 and SKR-1(SKP1). **B)** Scatter plot showing a single affinity purification proteomic experiment comparing proteins identified in GS::RPM-1 LD substrate 'trap' versus GS::RPM-1. Highlighted in red are CDK-5 and CDKA-1(p35) which are exclusive to GS::RPM-1 LD. Identified in both samples are known RPM-1 binding proteins (blue), components of the RPM-1 ubiquitin ligase complex (orange) and a previously validated substrate (UNC-51, orange). **C)** Scatter plot showing the same single proteomic experiment comparing GS::RPM-1 LD versus GS::GFP (negative control). Gray dashed line delineates two-fold enrichment. **D)** Example LC-MS/MS spectrum for one CDK-5 peptide identified in GS::RPM-1 LD sample. Shown are B ions formed from the amino terminus and Y ions formed from the carboxyl terminus. m/z represents mass to charge ratio with peptide size increasing to the right. **E)** Summary of results from 7 independent proteomic experiments with RPM-1. Significance determined using Student's *t*-test with p-values annotated. *ns*, not significant (p>0.05); NA, not applicable.

Previous studies showed that RPM-1 forms an atypical RING ubiquitin ligase complex with the F-box protein FSN-1 [32,33,36]. RPM-1 is the catalytic RING protein that transfers ubiquitin to substrates, while FSN-1 provides substrate recognition and specificity. We previously showed that FSN-1 binds directly to both RPM-1 and substrates, thereby recruiting substrates into this complex for ubiquitination by RPM-1 enzymatic activity [33,36].

We CRISPR engineered GFP or a 3xFLAG tag onto endogenous RPM-1, FSN-1 and CDK-5 (**Fig 2A**). RPM-1 was fused with GFP (GFP::RPM-1 CRISPR) at an internal location that we previously showed does not impair RPM-1 function [40]. We subsequently CRISPR edited three mutations (C3535A, H3537A, and H3540A) into the RING domain of GFP::RPM-1 to

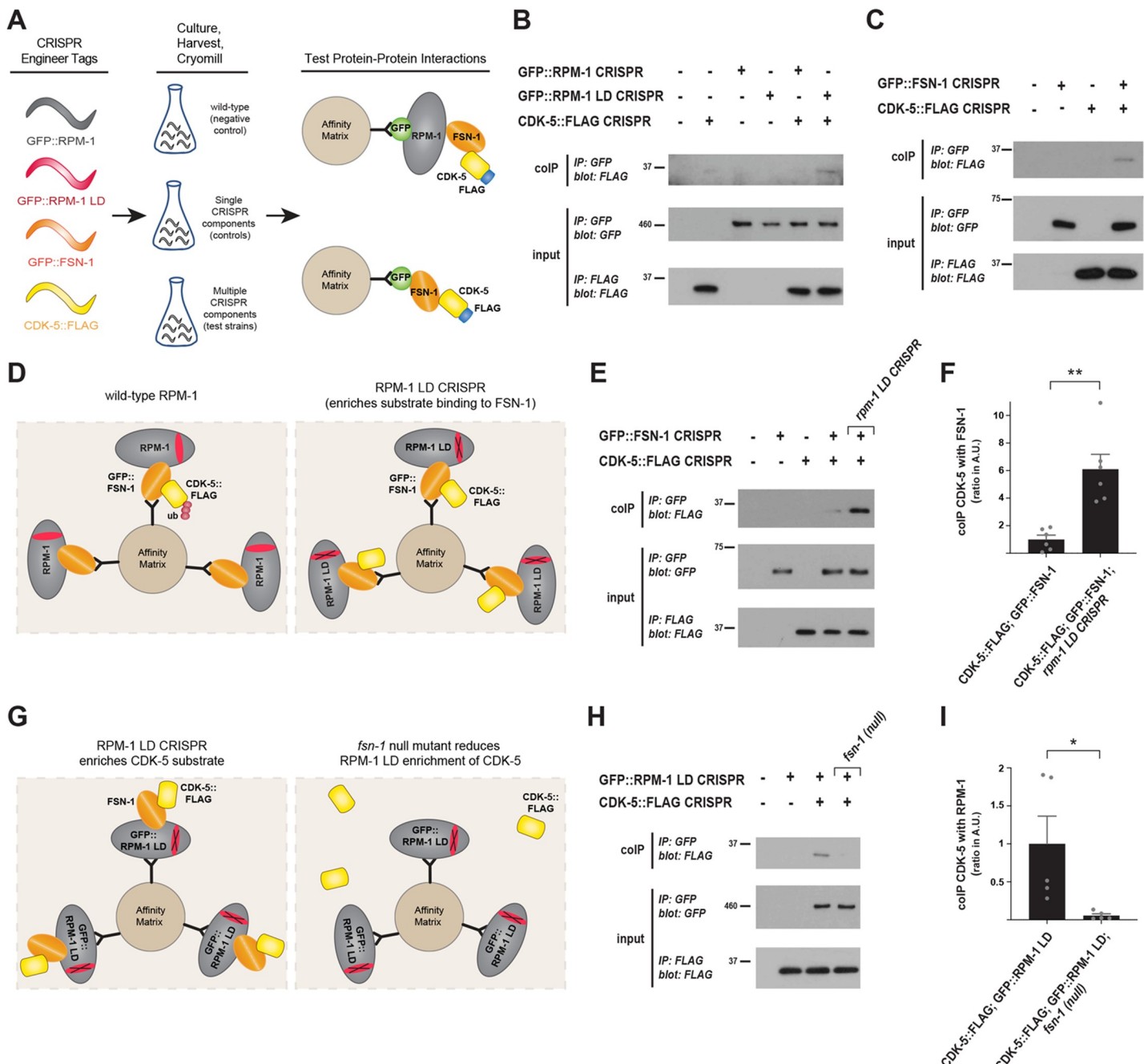

**Fig 2. CRISPR-based biochemistry demonstrates substrate-like interactions between CDK-5 and the RPM-1/FSN-1 ubiquitin ligase complex. A**) Schematic of experimental workflow for CRISPR-based, native biochemistry using *C. elegans*. Multiple rounds of CRISPR/Cas9 engineering was used to affinity tag and point mutate components of the RPM-1/FSN-1 ubiquitin ligase complex and CDK-5. **B**) CoIP showing increased binding of CDK-5::FLAG to GFP::RPM-1 LD substrate 'trap' compared to GFP::RPM-1. **C**) CoIP showing CDK-5::FLAG binds GFP::FSN-1. **D**) Schematic showing how *rpm-1 LD CRISPR* substrate 'trap' enriches CDK-5 binding to FSN-1, if CDK-5 is an RPM-1/FSN-1 substrate. **E**) CoIP showing increased binding between CDK-5::FLAG and GFP::FSN-1 in *rpm-1 LD CRISPR* animals (far right lane) compared to wildtype background (second lane from right). **F**) Quantification of CDK-5 coIP with FSN-1 in *rpm-1 LD CRISPR* animals compared to wildtype background. **G**) Schematic showing how RPM-1 LD will not bind to CDK-5 in the absence of FSN-1 substrate recognition module (*i.e. fsn-1* null mutant). **H**) CoIP showing reduced binding between CDK-5::FLAG and GFP::RPM-1 LD in *fsn-1 (null)* mutants (far right lane) compared to wildtype background (second lane from right). **I**) Quantification of CDK-5 coIP with RPM-1 LD in *fsn-1* null mutants compared to wildtype background. For B, C, and E, representative of three or more independent experiments is shown. For H, representative of two independent experiments is shown. For F and I, shown are means from 5–6 replicates from 3 or 2 independent experiments, respectively. Wildtype values were normalized to 1 (Arbitrary Units) to facilitate comparisons. Error bars indicate SEM. Significance assessed using Student's *t*-test. ** p<0.01.

render it ligase dead (GFP::RPM-1 LD CRISPR) [33,41]. Thus, we used CRISPR engineering to generate an endogenous GFP::RPM-1 LD substrate 'trap'. FSN-1 was N-terminally fused with GFP (GFP::FSN-1 CRISPR) and CDK-5 was C-terminally tagged with 3xFLAG (CDK-5:: FLAG CRISPR). These strains were crossed to obtain a variety of CRISPR engineered animals containing multiple tagged and catalytically altered components (Fig 2A). To test interactions, animals were ground to submicron particles under liquid $N_2$ temperatures using a cryomill, proteins were extracted in lysis buffer and co-immunoprecipitation (coIP) was performed.

We began by evaluating CDK-5 interactions with RPM-1 LD and wild-type RPM-1. Consistent with our proteomic results, we detected more binding of CDK-5::FLAG with GFP:: RPM-1 LD than with GFP::RPM-1 (Fig 2B). We then tested interactions between CDK-5 and the F-box protein FSN-1, which recognizes and binds ubiquitination substrates. CoIP results indicated that CDK-5::FLAG binds GFP::FSN-1 (Fig 2C).

Next, we tested whether the RPM-1 LD substrate 'trap' affects CDK-5 binding to FSN-1. To do so, we deployed native biochemistry with a three component CRISPR strain (Fig 2D). We used CRISPR editing to generate *rpm-1 LD CRISPR* mutants, which were then crossed with animals carrying GFP::FSN-1 CRISPR and CDK-5::FLAG CRISPR. CoIP was performed with these animals, control CRISPR engineered strains expressing single constructs, and wild-type animals. Interestingly, we observed a robust increase in binding of CDK-5::FLAG with GFP:: FSN-1 in *rpm-1 LD CRISPR* mutants (Fig 2E). Quantitation indicated that approximately 6-times more CDK-5::FLAG coprecipitates with GFP::FSN-1 in CDK-5::FLAG; GFP::FSN-1; *rpm-1 LD CRISPR* mutants compared to CDK-5::FLAG; GFP::FSN-1 CRISPR animals (Fig 2F).

To further probe CDK-5 interactions with the RPM-1/FSN-1 ubiquitin ligase complex, we tested whether removing the FSN-1 substrate recognition module affects binding between the RPM-1 LD substrate 'trap' and CDK-5 (Fig 2G). To do so, we generated GFP::RPM-1 LD CRISPR; CDK-5::FLAG CRISPR; *fsn-1* mutants using an *fsn-1* null allele. CoIP results indicated that RPM-1 LD binding to CDK-5 was impaired in the absence of FSN-1 (Fig 2H). Quantitation showed a significant reduction in binding between RPM-1 LD and CDK-5 in GFP::RPM-1 LD CRISPR; CDK-5::FLAG CRISPR; *fsn-1* mutants compared to GFP::RPM-1 LD CRISPR; CDK-5::FLAG CRISPR animals (Fig 2I).

In summary, we emphasize several points. 1) Using endogenous CRISPR engineered constructs with a further layer of CRISPR editing, we observe substrate-like interactions between CDK-5 and the RPM-1/FSN-1 complex in a native, whole organism setting. 2) The RPM-1 LD substrate 'trap' promotes the interaction between CDK-5 and the FSN-1 substrate recognition module. 3) Complementary results indicate that FSN-1 mediates CDK-5 binding to the RPM-1 LD substrate 'trap'. These outcomes with CRISPR-based native biochemistry are consistent with CDK-5 being a potential ubiquitination substrate for the RPM-1/FSN-1 ubiquitin ligase complex.

## RPM-1 and FSN-1 inhibit CDK-5 to regulate axon development

With proteomic and biochemical results suggesting that CDK-5 is an RPM-1 ubiquitination substrate, we next wanted to test the functional genetic relationship between *cdk-5* and *rpm-1*. To do so, we turned to the process of axon termination. Termination of axon growth has been used previously to evaluate the function of RPM-1 [42,43], its binding proteins [37–39], and its ubiquitination substrates which are known to be degraded by the proteasome [33,37,44]. Our prior work demonstrated that RPM-1 regulates axon termination by affecting growth cone collapse during development [42].

Axon termination was evaluated using the ALM mechanosensory neurons of *C. elegans*. There are two ALM neurons in *C. elegans* that have their somas in the mid-body of the animal

with each neuron extending a single axon toward the anterior. ALM axons terminate growth after the most anterior pharyngeal bulb (Fig 3A). Consistent with prior studies, we found that *rpm-1* (null) mutants have severe axon termination defects in which ALM axons extend into the tip of the nose and hook back towards the posterior (Fig 3B). Quantitation indicated that axon termination defects occur with high frequency in *rpm-1* mutants (Fig 3C). In contrast, ALM neurons of *cdk-5* loss-of-function (lf) mutants displayed normal axon termination similar to wild-type animals (Fig 3B and 3C). To test genetic interactions between *rpm-1* and *cdk-5*, we generated double mutants. Interestingly, we principally observed less severe axon termination defects or wild-type axon termination in *cdk-5; rpm-1* double mutants (Fig 3B). Quantitation indicated that the frequency of axon termination defects was partially but significantly

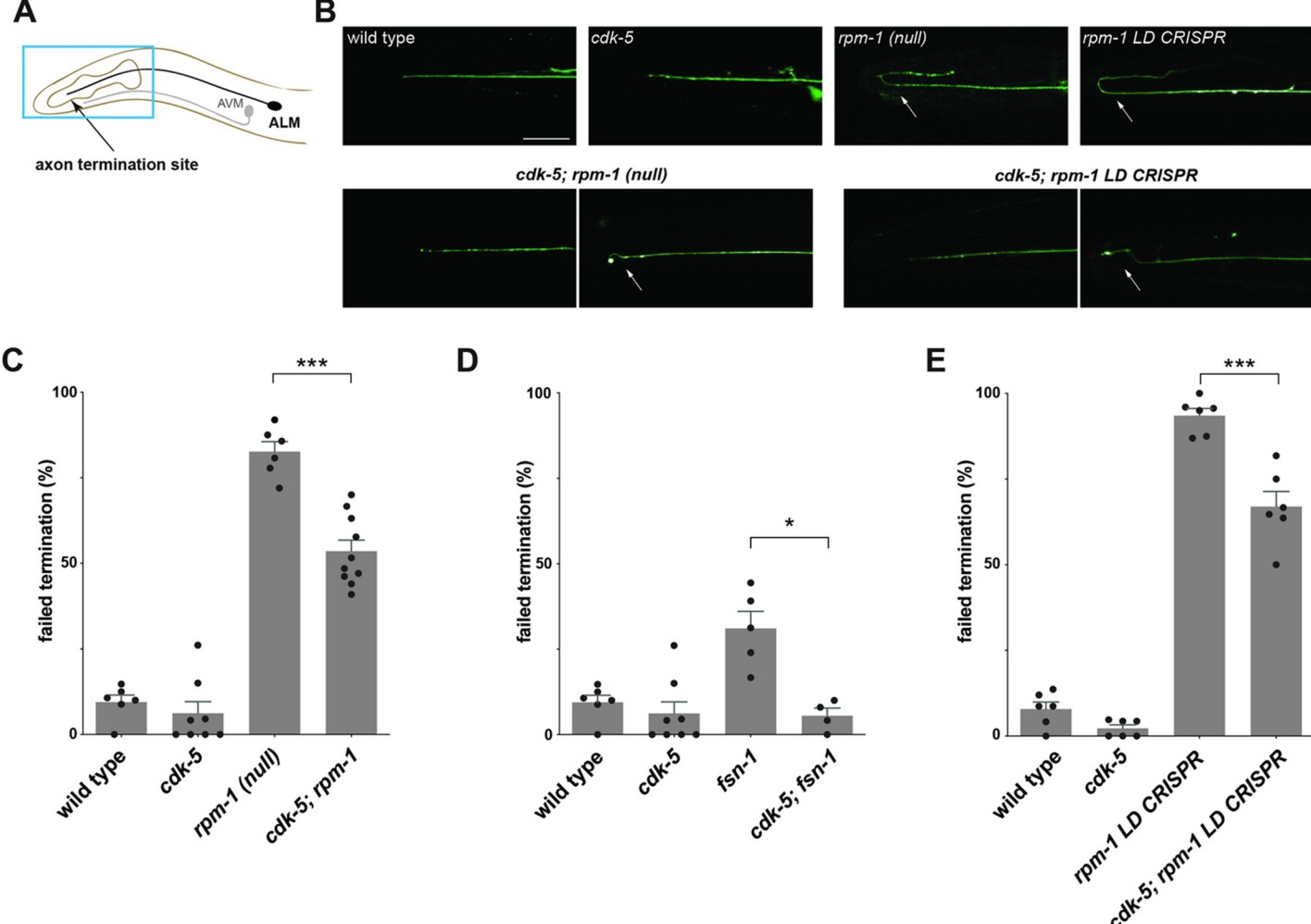

**Fig 3. RPM-1/FSN-1 ubiquitin ligase complex activity inhibits CDK-5 to regulate axon termination in *C. elegans*. A)** Schematic shows axon termination sites for anterior mechanosensory neurons, ALM and AVM, of *C. elegans*. **B)** Representative images of axon termination in ALM neurons for indicated genotypes visualized using transgenic GFP (*muIs32*). Shown are severe axon termination defects in *rpm-1* (lf) mutants and *rpm-1 LD CRISPR* animals (arrows). Axon termination defects were not observed in *cdk-5* single mutants. In *cdk-5; rpm-1* double mutants and *cdk-5; rpm-1 LD CRISPR* double mutants two examples are shown for wild-type axon termination (left) and mild axon termination defects (right, arrows) which are both observed in these animals. **C-E)** Quantitation of axon termination defects in ALM neurons for indicated genotypes. Axon termination defects are reduced in **C)** *cdk-5; rpm-1* double mutants, **D)** *cdk-5; fsn-1* double mutants, and **E)** *cdk-5; rpm-1 LD CRISPR* double mutants compared to single mutants. For C-E, means (bars) are shown for 4 or more counts (black dots represent individual counts, 20 or more worms/count) for each genotype. Error bars indicate SEM. Significance assessed using Student's *t*-test with Bonferroni correction. *** p<0.001, * p<0.05. Scale bars 20 μm.

suppressed in *cdk-5; rpm-1* double mutants compared to *rpm-1* single mutants (**Fig 3C**). Importantly, our genetic analysis was performed using an *rpm-1* null allele, *ju44*, and a likely null allele for *cdk-5*, *ok626* [45,46]. Thus, suppression of axon termination defects in *cdk-5; rpm-1* double mutants suggests that RPM-1 functions as an upstream inhibitor of CDK-5.

Next, we tested the functional relationship between the F-box protein FSN-1 and CDK-5. Like prior studies [37,39], we observed axon termination defects in ALM neurons of *fsn-1* null mutants (**Fig 3D**). Axon termination defects were significantly suppressed in *cdk-5; fsn-1* double mutants compared to *fsn-1* single mutants (**Fig 3D**). These findings indicate that FSN-1, like RPM-1, regulates ALM axon termination by inhibiting CDK-5.

Because CDK-5 is activated by CDKA-1(p35), we tested whether *cdka-1* shows a similar genetic relationship with *rpm-1*. Like *cdk-5*, *cdka-1* single mutants displayed normal axon termination, and axon termination defects were suppressed in *cdka-1; rpm-1* double mutants compared to *rpm-1* single mutants (**S2 Fig**).

Previous studies in *C. elegans* have shown that the PCT-1 kinase can function redundantly with CDK-5 [47]. Therefore, we asked whether *pct-1* interacts genetically with *rpm-1*. Unlike *cdk-5; rpm-1* double mutants, we did not observe suppression of axon termination defects in *pct-1; rpm-1* double mutants generated using a *pct-1* null allele (**S3 Fig**). Moreover, suppression was not enhanced in *cdk-5; pct-1; rpm-1* triple mutants beyond the level of suppression observed in *cdk-5; rpm-1* double mutants (**S3 Fig**). These results indicate that CDK-5 and PCT-1 do not function redundantly during axon termination.

Taken as a whole, these findings demonstrate that ALM mechanosensory neurons are a suitable readout for evaluating genetic interactions between *cdk-5*, *rpm-1* and *fsn-1*. Furthermore, genetic outcomes suggest that the RPM-1/FSN-1 ubiquitin ligase complex inhibits CDK-5 to control axon termination *in vivo*.

## RPM-1 ubiquitin ligase activity inhibits CDK-5

To take our genetic studies a step further, we wanted to determine if *cdk-5* (lf) suppresses defects caused by specifically impairing RPM-1 ubiquitin ligase activity. To test this, we turned to *rpm-1 LD CRISPR* animals where point mutations in the RING domain disrupt zinc binding and impair the transfer of ubiquitin to substrates [33,41]. We observed severe axon termination defects in *rpm-1 LD CRISPR* mutants (**Fig 3B**). Quantitation indicated that axon termination defects occur at high frequency (**Fig 3E**). In *cdk-5; rpm-1 LD CRISPR* double mutants, we observed less severe axon termination defects or wild-type axon termination events (**Fig 3B**). Quantitation showed that axon termination defects were partially but significantly suppressed in *cdk-5; rpm-1 LD CRISPR* double mutants compared to *rpm-1 LD CRISPR* animals (**Fig 3E**). These findings demonstrate that RPM-1 ubiquitin ligase activity regulates axon termination by inhibiting CDK-5.

## CDK-5 and RPM-1 have opposing effects on axon development

Two more approaches were used to test our model that the RPM-1/FSN-1 ubiquitin ligase complex regulates axon termination by inhibiting CDK-5 (**Fig 4A**). 1) We used transgenic rescue experiments to verify genetic suppressor effects, and to test if RPM-1 inhibits CDK-5 in mechanosensory neurons. 2) If the RPM-1/FSN-1 ubiquitin ligase complex inhibits CDK-5, then excess CDK-5 activity would impair axon termination. We used transgenic overexpression experiments to test this hypothesis.

For transgenic rescue experiments, we expressed CDK-5 in *cdk-5; rpm-1* double mutants and evaluated axon termination in ALM mechanosensory neurons. We generated extrachromosomal transgenic arrays that express the genomic *cdk-5* locus including native promoter

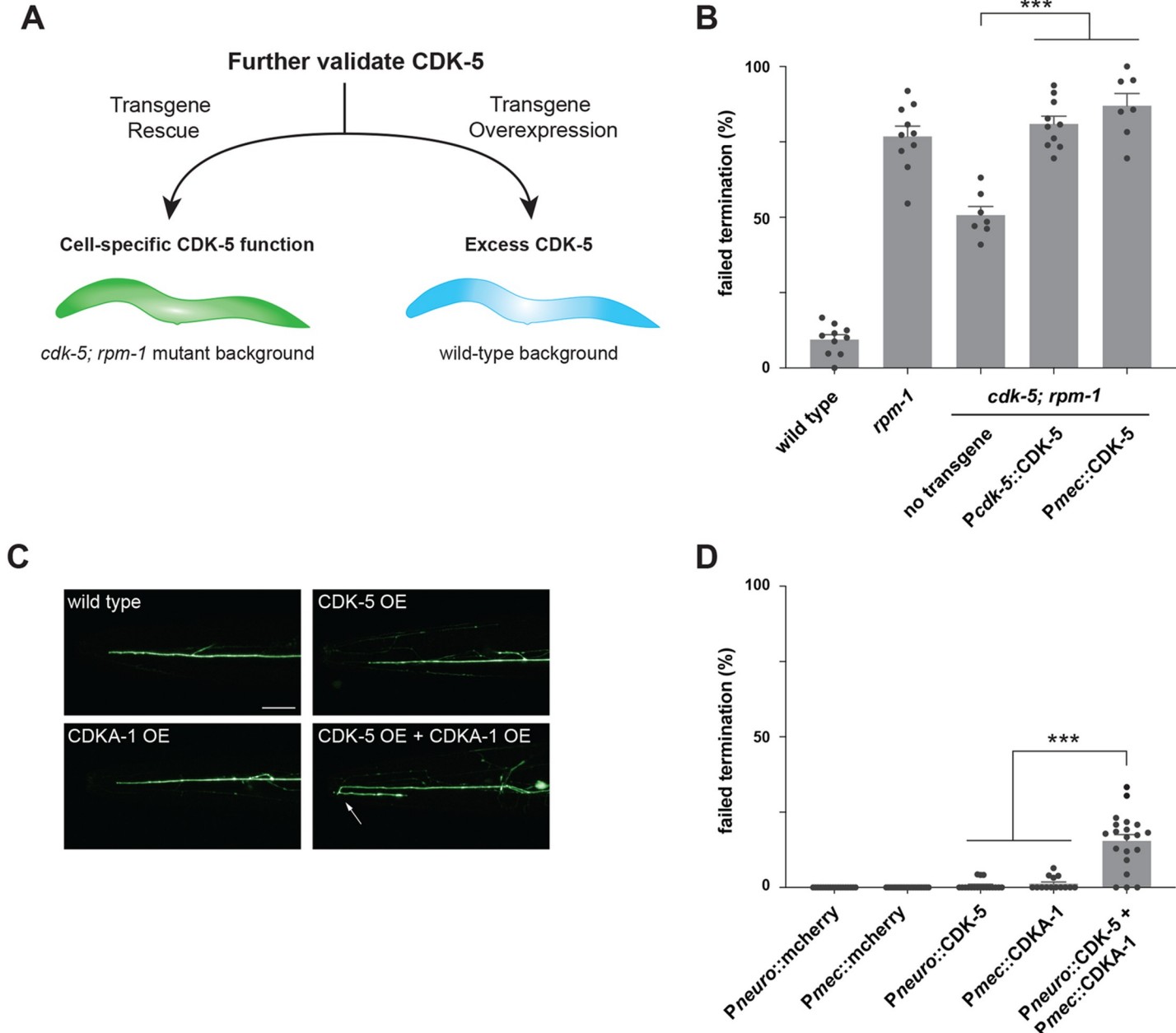

**Fig 4. Transgenic rescue and overexpression studies indicate CDK-5 functions cell-autonomously to regulate axon termination. A**) Schematic of transgenic workflow for evaluating CDK-5 rescue, cell-autonomous function and overexpression. **B**) Quantitation of axon termination defects in ALM neurons indicates transgenic CDK-5 expressed using native promoter (P*cdk-5*) or promoter for mechanosensory neurons (P*mec*) rescues reduced axon termination defects in *cdk-5; rpm-1* double mutants. **C**) Representative images of axon termination in ALM neurons of transgenic animals overexpressing indicated constructs. Axon termination is impaired with overexpression of both CDK-5 and CDKA-1(p35). **D**) Quantitation of ALM axon termination defects caused by transgenic overexpression of indicated constructs. For all experiments, means (bars) are shown from data collected from 3 or more independent transgenic lines (two counts/line, black dots) for each genotype. Error bars indicate SEM. Significance assessed using Student's *t*-test with Bonferroni correction. *** p<0.001. Scale bars 20 μm.

and 3' UTR (P*cdk-5*::CDK-5). Expression of P*cdk-5*::CDK-5 completely rescued suppression in *cdk-5; rpm-1* double mutants causing axon termination defects to return to levels observed in *rpm-1* single mutants (**Fig 4B**). Next, we transgenically expressed *cdk-5* cDNA using a promoter that is specific for mechanosensory neurons (P*mec*::CDK-5) in *cdk-5; rpm-1* double

mutants. Once again, we observed robust, significant rescue of suppression (**Fig 4B**). These results with both native and mechanosensory neuron promoters confirm that suppression in *cdk-5; rpm-1* double mutants is caused by loss of CDK-5. Furthermore, our findings demonstrate that CDK-5 affects axon termination by functioning cell-autonomously in mechanosensory neurons.

Because multiple genetic outcomes indicated that RPM-1 inhibits CDK-5, it is plausible that excess CDK-5 activity occurs in the absence of RPM-1 leading to defective axon termination (**Fig 4A**). To test this, we generated animals with transgenic extrachromosomal arrays that use a pan-neuronal promoter to overexpress CDK-5 (P*neuro*::CDK-5). However, overexpression of CDK-5 alone did not impair axon termination (**Fig 4C and 4D**). Because the activity of excess CDK-5 might be limited by endogenous levels of its activator, CDKA-1(p35), we generated transgenic animals that overexpressed both CDK-5 and CDKA-1. We again used a pan-neuronal promoter to overexpress CDK-5, and used a mechanosensory neuron promotor to overexpress CDKA-1 (P*mec*::CDKA-1). Simultaneously overexpressing both components resulted in significant axon termination defects in ALM neurons (**Fig 4C and 4D**). We further evaluated several controls in which each promoter drove mCherry (negative control) or CDKA-1 alone. None of these transgenic controls yielded axon termination defects (**Fig 4D**). Thus, simultaneous overexpression of both CDK-5 and its activator CDKA-1 is needed to impair axon termination. Importantly, this is a similar phenotype to *rpm-1* (lf) mutants (**Fig 3**). It is reasonable that CDK-5/CDKA-1 overexpression did not result in the same high frequency defects as *rpm-1* (lf) mutants, because RPM-1 functions as both a signaling hub and ubiquitin ligase that influences several downstream pathways and ubiquitination substrates [30]. More modest CDK-5 overexpression results (**Fig 4D**) are also consistent with *cdk-5* loss of function partially suppressing axon termination defects caused by *rpm-1* loss of function (**Fig 3C and 3E**). Indeed, partial suppression of *rpm-1* mutant phenotypes has been observed for other validated RPM-1 substrates [33].

Overall, our results support two conclusions. First, CDK-5 functions cell autonomously in mechanosensory neurons to regulate axon termination. Second, overexpression of both CDK-5 and CDKA-1 is sufficient to impair axon termination in ALM neurons. Thus, CDK-5 gain-of-function effects generate a similar phenotype to *rpm-1* (lf). These findings provide further evidence indicating that RPM-1 inhibits CDK-5 to regulate axon termination in mechanosensory neurons.

## RPM-1 restricts CDK-5 kinase activity

Next, we wanted to determine if RPM-1 specifically restrains CDK-5 kinase activity. We addressed this with two experimental paradigms (**Fig 5A**). In the first, we used CRISPR/Cas9 editing to create a *cdk-5* kinase dead (KD) allele, which we evaluated for genetic interactions with *rpm-1*. Second, we utilized cell-specific transgenics to test whether CDK-5 kinase activity is required in mechanosensory neurons for axon termination.

To catalytically inactivate CDK-5 and create a CDK-5 KD allele, we CRISPR mutated two conserved amino acids: K33T which impairs ATP binding and D144N which is required for phosphate transfer to substrates (**Fig 5A**) [48]. K33 and D144 are identical residues in *C. elegans*, fly and mammalian CDK-5 and part of highly conserved motifs (**Fig 5A**). *cdk-5 KD CRISPR* mutants showed no defects in axon termination of ALM neurons (**Fig 5B**). *cdk-5 KD CRISPR*; *rpm-1* double mutants showed a significant reduction in axon termination defects compared to *rpm-1* single mutants (**Fig 5B**). Thus, the *cdk-5 KD CRISPR* allele suppresses *rpm-1* (lf) defects similar to the *cdk-5* null allele (**Fig 3B**)

For cell-specific transgenics, we used a mechanosensory neuron promoter to express a CDK-5 KD cDNA (P*mec*::CDK-5 KD). As a positive control, we evaluated transgenic

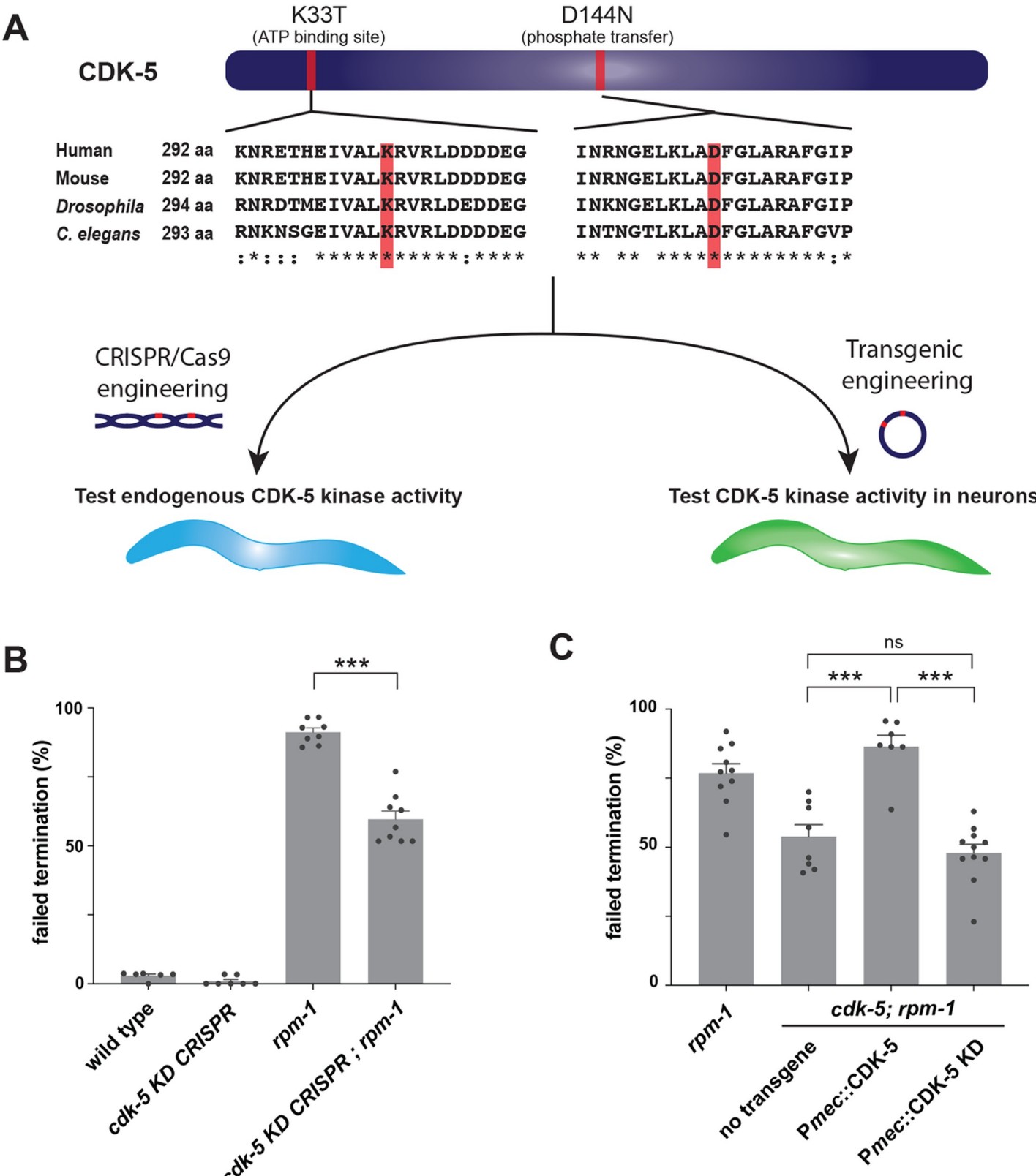

**Fig 5. CRISPR editing and transgenics demonstrate RPM-1 restricts CDK-5 kinase activity. A**) Protein sequence alignment showing two conserved CDK-5 residues (red) mutated to inactivate kinase activity. Workflow for evaluating CDK-5 kinase activity using CRISPR/Cas9 editing or transgenic engineering. **B**) CRISPR evaluation

of CDK-5 kinase activity. Quantitation indicates axon termination defects in ALM neurons are reduced in *rpm-1* mutants when CDK-5 is inactivated by CRISPR (*cdk-5* KD CRISPR; *rpm-1*). **C**) Transgenic evaluation of CDK-5 kinase activity. Quantitation indicates transgenic expression of CDK-5 KD in mechanosensory neurons fails to rescue reduced axon termination defects in ALM neurons of *cdk-5; rpm-1* double mutants. For all experiments, means (bars) are shown from data collected from 3 or more independent transgenic lines (two counts/line, black dots) for each genotype. Error bars indicate SEM. Significance assessed using Student's *t*-test with Bonferroni correction. ns, non-significant; *** p<0.001.

expression of wild-type CDK-5 in mechanosensory neurons (P*mec*::CDK-5). Once again, we observed that wild-type CDK-5 expression robustly rescued suppression in *cdk-5; rpm-1* double mutants with defects returning to a similar frequency as *rpm-1* single mutants (**Fig 5C**). In contrast, expression of CDK-5 KD in mechanosensory neurons failed to rescue suppression in *cdk-5; rpm-1* double mutants (**Fig 5C**).

Thus, results with CRISPR gene editing and transgenic rescue experiments show that RPM-1 restricts CDK-5 kinase activity. Moreover, CDK-5 kinase activity needs to be restrained in a cell-autonomous manner for ALM neurons to terminate axon growth accurately.

## CDK-5 and RPM-1 colocalize in the neuronal soma

Because CDK-5 showed substrate-like interactions with the RPM-1/FSN-1 ubiquitin ligase complex, we asked where their physical interaction might occur. To address this, we CRISPR engineered an mScarlet tag onto endogenous CDK-5 (CDK-5::mScarlet CRISPR). We observed widespread expression of CDK-5 throughout the animal's body including in the nervous system. We then combined CDK-5::mScarlet CRISPR with transgenic, cell-specific GFP to accurately determine if CDK-5 is expressed in ALM neurons. Indeed, we observed CDK-5:: mScarlet in ALM neurons, where it was principally localized to the neuronal soma (**Fig 6A**

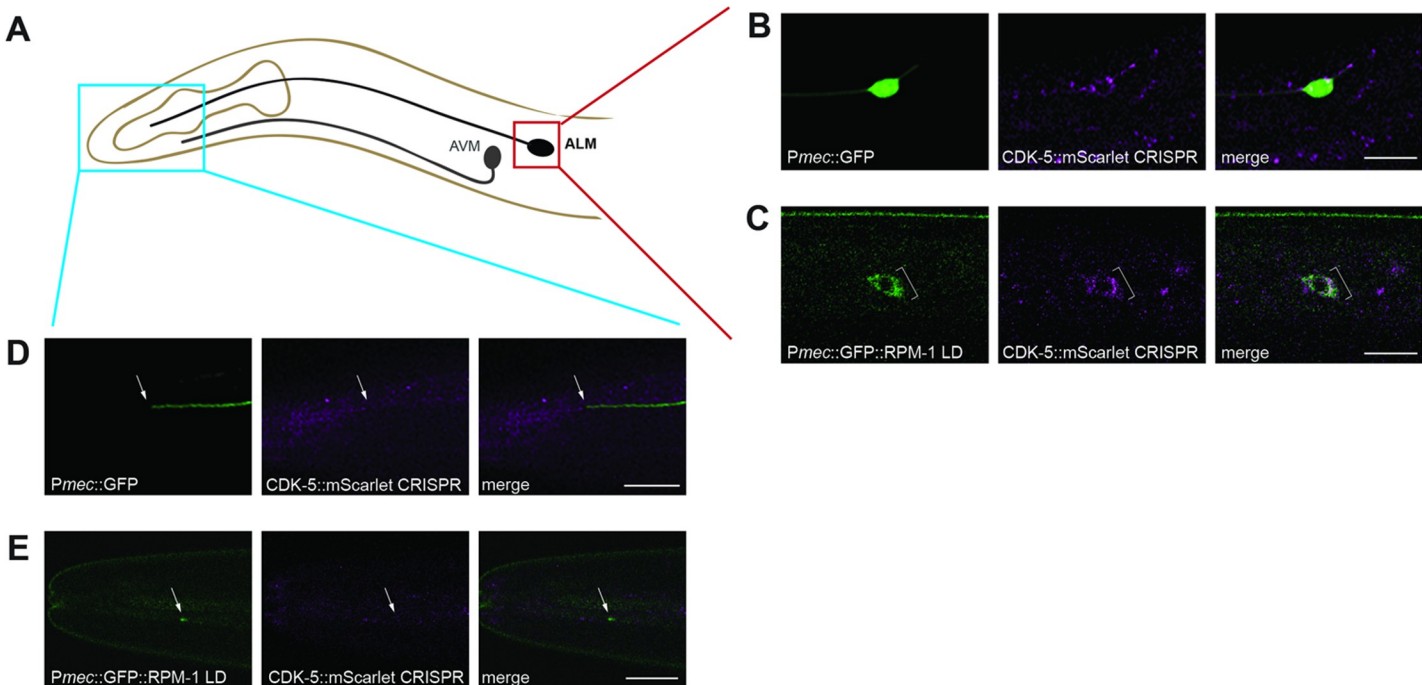

**Fig 6. CDK-5 colocalizes with RPM-1 LD in neuronal soma. A**) Schematic showing ALM mechanosensory neuron and region imaged for axon termination site (blue) and soma (red). **B**) CDK-5::mScarlet CRISPR is present in the soma of ALM neurons (bracket) visualized using transgenic GFP expressed in mechanosensory neurons (*muIs32*). **C**) CDK-5::mScarlet colocalizes with GFP::RPM-1 LD in ALM soma (bracket). **D**) Representative images showing CDK-5::mScarlet is not detected at the ALM axon tip visualized using transgenic GFP (arrow). **E**) GFP::RPM-1 LD accumulates at the ALM axon tip (arrow) where CDK-5::mScarlet is not detected. Scale bars 10 μm.

and 6B). We did not detect CDK-5 at the tip of the ALM axon (Fig 6A and 6D). While this suggests that CDK-5 could be absent from the axon, we cannot rule out that low sensitivity of endogenous CDK-5::mScarlet CRISPR is preventing us from visualizing axonal CDK-5.

To determine if the RPM-1 LD substrate 'trap' localizes with CDK-5 in similar cellular compartments, we transgenically expressed GFP::RPM-1 LD specifically in the mechanosensory neurons of CDK-5::mScarlet CRISPR animals. Previous studies showed that RPM-1 LD and wild-type RPM-1 have similar subcellular localization patterns in mechanosensory neurons [33]. We opted to use the RPM-1 LD because it could facilitate co-compartmentalization with CDK-5::mScarlet CRISPR. Indeed, we did observe GFP::RPM-1 LD colocalizing with and adjacent to CDK-5::mScarlet in the ALM soma (Fig 6C). Colocalization was most noticeable in perinuclear puncta. In contrast, CDK-5 was not detected at the terminated axon tip where RPM-1 also concentrates (Fig 6E). These results suggest that RPM-1 interacts with and inhibits CDK-5 in the neuronal soma.

Inhibition of CDK-5 by RPM-1/FSN-1 ubiquitin ligase complex suggests that CDK-5 ubiquitination could result in degradation by the proteasome. Therefore, we evaluated the level of CDK-5::mScarlet CRISPR in the ALM soma of adult wild-type animals compared to *rpm-1* (lf) mutants. Quantitative results indicate that CDK-5::mScarlet levels were not significantly different between wild-type animals and *rpm-1 (lf)* mutants (S4 Fig). There are several explanations for why we might not observe a change in CDK-5 levels in *rpm-1* mutants. 1) It is possible that the RPM-1/FSN-1 complex targets only a small portion of CDK-5 for ubiquitination and degradation by the proteasome. 2) The RPM-1/FSN-1 complex might inhibit CDK-5 via ubiquitination but not proteasome-mediated degradation. 3) Finally, we analyzed adult animals in our study that have completed the axon termination process. Thus, it remains possible that the RPM-1/FSN-1 complex restricts CDK-5 during development and that changes in CDK-5 levels are no longer detected in adult *rpm-1* mutants.

## RPM-1 inhibits CDK-5 to regulate axon development in SAB motor neurons

Our results with ALM mechanosensory neurons indicate that RPM-1 functions as an upstream inhibitor of CDK-5. To determine if this is a broad functional relationship in the nervous system or specific to ALM neurons, we tested genetic interactions between *cdk-5* and *rpm-1* using the SAB motor neurons as an independent readout for axon development.

There are 3 SAB motor neurons located in the head of *C. elegans*. Their cell bodies are near the second pharyngeal bulb of the animal, and they extend their axons anteriorly towards the nose (Fig 7A). Consistent with prior studies [43], we observed axon termination defects in the SAB neurons of *rpm-1* (lf) mutants (Fig 7A). Quantitation showed that SAB axon termination defects were significant and occurred at moderate frequency in *rpm-1* mutants (Fig 7B). In contrast, axon termination was not impaired in the SAB neurons of *cdk-5* mutants (Fig 7A and 7B). We observed both defective and wild-type SAB axon termination in *cdk-5; rpm-1* double mutants (Fig 7A). Quantitative results showed that SAB axon termination defects were partially but significantly suppressed in *cdk-5; rpm-1* double mutants compared to *rpm-1* single mutants (Fig 7B).

These genetic outcomes with null alleles demonstrate that RPM-1 inhibits CDK-5 to regulate axon termination in SAB motor neurons. Thus, findings with two different types of neurons, SAB motor neurons (Fig 7) and ALM mechanosensory neurons (Fig 3), demonstrate that RPM-1 functions as an inhibitory genetic mechanism that restricts CDK-5 *in vivo*.

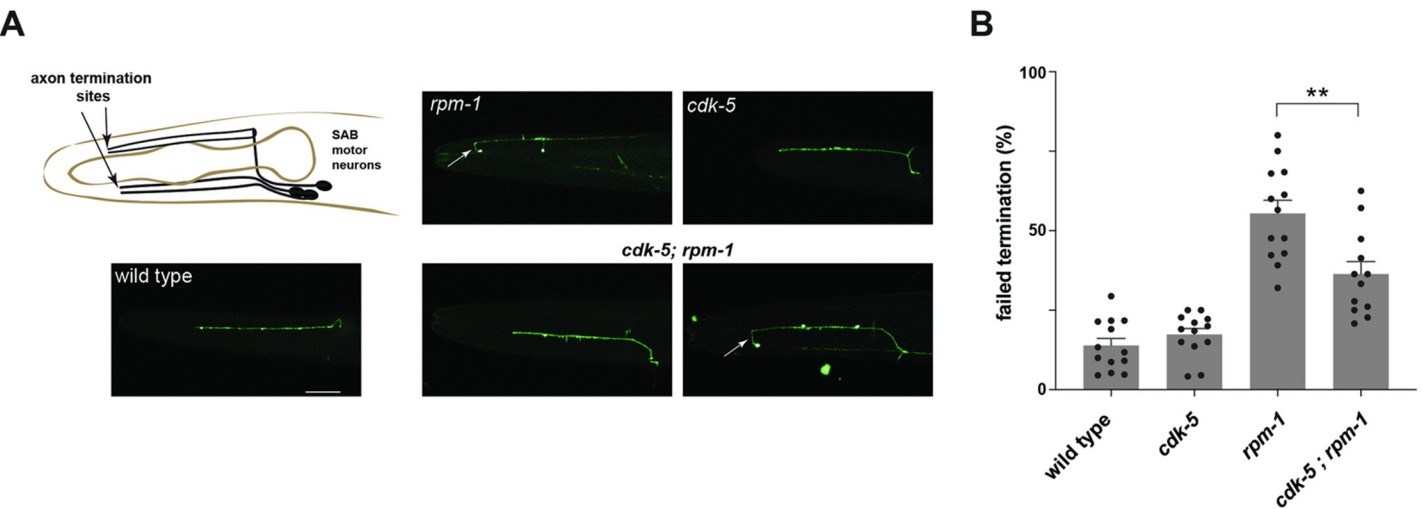

**Fig 7. RPM-1 inhibits CDK-5 to regulate axon termination in SAB motor neurons. A)** Schematic shows axon termination sites for SAB neurons of *C. elegans*. Representative images of SAB termination for indicated genotypes visualized using transgenic GFP (*wdIs4*). Shown is an example of axon termination defects in *rpm-1* (null) mutants (arrow). Axon termination defects were not observed in *cdk-5* single mutants. For *cdk-5; rpm-1* double mutants, two examples are shown for wild-type (left) and defective axon termination (right, arrow) which are both observed in these animals. **B)** Quantitation indicates axon termination defects in SAB neurons are reduced in *cdk-5; rpm-1* double mutants compared to *rpm-1* single mutants. Shown are means (bars) for 12 or more counts (black dots, 20 or more worms/count) for each genotype. Error bars indicate SEM. Significance assessed using Student's *t*-test with Bonferroni correction. ** p<0.01. Scale bars 20 μm.

## RPM-1 and CDK-5 act in parallel to control synapse formation

RPM-1 regulates not only axon termination but also synapse maintenance in mechanosensory neurons [43,49]. Likewise, CDK-5 has been shown to have a role in forming new synapses during developmental synapse remodeling in DD motor neurons [50]. Therefore, we examined the functional relationship between *rpm-1* and *cdk-5* in synapse formation using ALM mechanosensory neurons.

Each ALM neuron extends a single collateral branch from the primary axon and forms *en passant* synapses with interneurons in the nerve ring of *C. elegans* (**S5A Fig**). To evaluate synapse formation in mechanosensory neurons, we used cell-specific transgenes expressing GFP:: RAB-3 to label presynaptic terminals and RFP to visualize axon/branch architecture (**S5B Fig**). Confirming prior findings [43], we found that ALM neurons of *rpm-1* null mutants have synapse formation defects (**S5B Fig**). Quantitation indicated that the number of presynaptic RAB-3 puncta are significantly reduced in *rpm-1* mutants compared to wild-type animals (**S5C Fig**). We did not observe a defect in RAB-3 puncta numbers in *cdk-5* mutants (**S5B and S5C Fig**). However, *cdk-5; rpm-1* double mutants displayed a more severe phenotype with fewer RAB-3 puncta than *rpm-1* single mutants (**S5B Fig**). Quantitation indicated that *cdk-5; rpm-1* double mutants have enhanced defects in RAB-3 puncta compared to *rpm-1* single mutants (**S5C Fig**). These results suggest that RPM-1 and CDK-5 act in parallel pathways to regulate synapse formation in ALM neurons. Our findings are consistent with prior studies that showed RPM-1 and CDK-5 affect different steps in the synapse formation process.

## Discussion

Cdk5 regulates numerous functions in the nervous system, is involved in neurodegenerative disease, and affects cancer progression [1,2,4]. Like Cdk5, ubiquitin ligases are prominent players in nervous system development, function and disease, and have important roles in cancer [26,27,51]. Here, we show that Cdk5 is regulated by ubiquitin ligase activity in the nervous

system. We deploy several approaches in *C. elegans* to draw this conclusion including proteomics, CRISPR-based biochemistry, traditional genetics and CRISPR gene editing. Collectively, our results indicate that the RPM-1/FSN-1 ubiquitin ligase complex inhibits CDK-5 to control axon termination *in vivo*. Our finding that ubiquitin ligase activity restricts Cdk5 in the nervous system could have implications for how Cdk5 is regulated in other cellular settings and disease states.

## CDK-5 is negatively regulated by the RPM-1/FSN-1 ubiquitin ligase complex

Previous studies identified cyclin E, cyclin D1 and GSTP1 as negative regulators of Cdk5 [21–23]. These inhibitors share a similar mechanism whereby they compete with p35 and p39 activators for binding to Cdk5. Here, we identify the RPM-1/FSN-1 ubiquitin ligase complex as a new inhibitory mechanism that restricts CDK-5 in the nervous system (**Fig 8**).

Previous studies established that RPM-1 and FSN-1 form a ubiquitin ligase complex that is orthologous to the complex formed by human MYCBP2(PAM) and FBXO45 [33–36]. RPM-1 is a RING ubiquitin ligase that transfers ubiquitin to substrates, and FSN-1 is the F-box protein that provides specificity by binding directly to both the substrate and RPM-1. Unlike canonical Cullin RING Ligase complexes (CRL), RPM-1 forms a non-canonical RING ubiquitin ligase complex that contains an F-box protein (FSN-1) and SKP1 (SKR-1) but lacks a Cullin

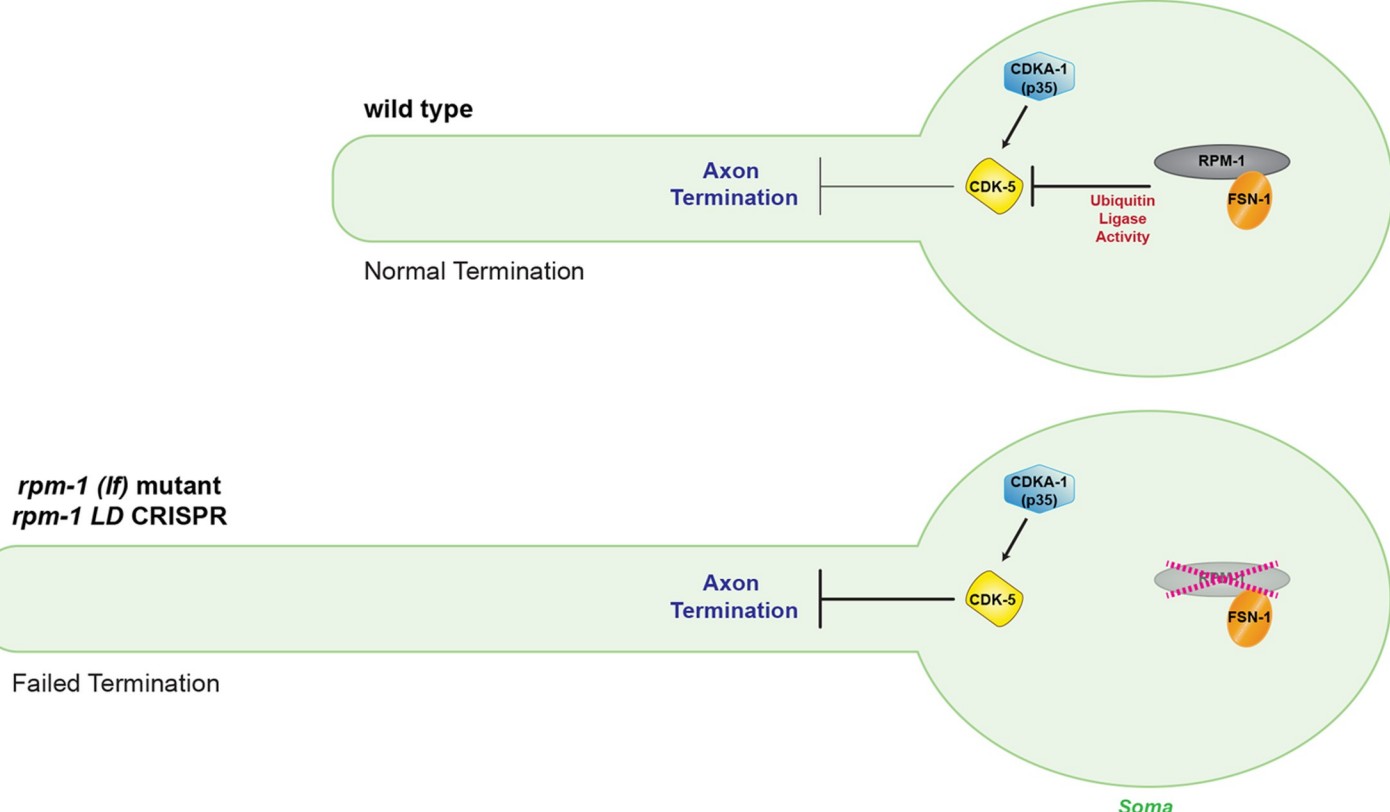

**Fig 8. RPM-1/FSN-1 ubiquitin ligase complex inhibits CDK-5 to control axon termination.** Summary showing CDK-5 is restricted by RPM-1/FSN-1 ubiquitin ligase activity in the neuronal soma. In wild-type animals (upper diagram), the RPM-1/FSN-1 ubiquitin ligase complex inhibits CDK-5 to facilitate axon termination. In *rpm-1* (lf) mutants and *rpm-1 LD CRISPR* animals (lower diagram), the activity of the RPM-1/FSN-1 ubiquitin ligase complex is impaired and CDK-5 is not restrained. As a result, excess CDK-5 kinase activity impairs axon termination resulting in excess axon growth. CDKA-1(p35) is a known CDK-5 activator.

[33,36,52]. RPM-1 is also an atypical RING-Cysteine-Relay (RCR) ligase that can ubiquitinate residues other than lysine, such as threonine [32]. To date, all known RPM-1 substrates are ubiquitinated and degraded by the 26S proteasome [36,44,53–55].

Our proteomic and biochemical studies revealed a series of substrate-like interactions between the RPM-1/FSN-1 ubiquitin ligase complex and CDK-5. *In vivo* affinity purification proteomics from *C. elegans* using an RPM-1 LD substrate 'trap' identified CDK-5 as a putative substrate (**Fig 1**). Importantly, this RPM-1 substrate 'trap' was previously shown to enrich a verified, functionally validated substrate [33]. Physical interactions between CDK-5 and both RPM-1 and FSN-1 were demonstrated using CRISPR-based native biochemistry (**Fig 2**). The RPM-1 LD substrate 'trap' increased binding between CDK-5 and the FSN-1 substrate recognition module (**Fig 2**). Furthermore, deleting endogenous FSN-1 abolished the interaction between CDK-5 and RPM-1 LD (**Fig 2**). Thus, the RPM-1/FSN-1 ubiquitin ligase complex displays a series of substrate-like interactions with CDK-5 in an endogenous, whole animal setting.

Our genetic findings indicate that RPM-1/FSN-1 ubiquitin ligase activity promotes accurate termination of axon growth by inhibiting CDK-5 (**Fig 8**). This is based on our observation that *cdk-5* (lf) genetically suppresses axon termination defects caused by *rpm-1* (lf), *fsn-1* (lf) and *rpm-1 LD CRISPR* animals that specifically lack RPM-1 ubiquitin ligase activity (**Fig 3**). Consistent with these findings, overexpressing CDK-5 with its activator CDKA-1 (**Fig 4**) impaired axon termination similar to *rpm-1* (lf) and *fsn-1* (lf) (**Fig 3**). These inhibitory genetic relationships are similar to UNC-51/ULK and DLK-1 which are ubiquitinated by RPM-1 and targeted for proteasome-mediated degradation [33,44]. While it remains unclear if RPM-1/FSN-1 inhibition of CDK-5 occurs via the proteasome, this is a reasonable possibility. However, our findings do not rule out the alternative of proteasome-independent inhibition.

Our study is not alone in indicating that ubiquitin ligase activity can regulate Cdk5. Prior cell-based experiments with overexpressed proteins showed that Cdk5 can be ubiquitinated, and that the Cdh1 ubiquitin ligase is sufficient to trigger proteasome-mediated turnover of Cdk5 [29]. We have now identified the RPM-1/FSN-1 ubiquitin ligase complex as an inhibitor of Cdk5 *in vivo* and in the nervous system. Our discovery and this prior work support the intriguing possibility that there could be several ubiquitin ligases that restrict Cdk5. Moreover, the RPM-1/FSN-1 ubiquitin ligase complex is potentially part of a larger regulatory network that inhibits Cdk5 and its activators. Such a network might include the ubiquitin ligases PJA2 and Mind bomb 1 (MIB1) that regulate p35 [56–59], as well as molecules such as cyclin E that inhibit p35 binding to Cdk5 [21]. Further evidence for and evaluation of a possible negative regulatory network that restricts Cdk5 now awaits further investigation.

## Axon development requires inhibition of CDK-5

Cdk5 is an important regulator of axon and neurite outgrowth [48,60–62]. The tyrosine kinases Abl and Fyn positively regulate Cdk5 during neurite outgrowth and growth cone collapse [13,14]. Cdk5 affects axon development and growth cone collapse by phosphorylating several molecules including Axin, LMTK1 and Doublecortin [63–65]. Despite over two decades of study, the inhibitory mechanisms that regulate Cdk5 during axon and neurite development have remained elusive.

In this study, we show that the RPM-1/FSN-1 ubiquitin ligase complex restrains CDK-5 kinase activity to regulate axon development in *C. elegans*. Previous work using Drosophila and mice indicated that Cdk5 can affect axon guidance [60,66,67]. Here, we show that RPM-1/FSN-1 ubiquitin ligase activity restricts CDK-5 to promote axon termination. In the developmental program, axon termination is a later step than axon guidance, and proper axon termination is essential for accurate, efficient nervous system construction [42,68]. Regarding

synapse formation, our less-comprehensive genetic findings suggest that RPM-1 and CDK-5 could have a different functional relationship in this context (**S4 Fig**). Why this is the case remains unclear but could be because RPM-1 and CDK-5 are known to affect different steps in the synapse formation process [49,50].

We evaluated axon termination in *C. elegans* using two types of neurons, ALM mechanosensory neurons (**Fig 3**) and SAB motor neurons (**Fig 7**). Genetic results using traditional loss of function alleles and CRISPR editing showed that RPM-1/FSN-1 ubiquitin ligase activity inhibits CDK-5 to facilitate axon termination. We further used CRISPR editing and transgenic rescue experiments to demonstrate that CDK-5 kinase activity functions cell-autonomously in mechanosensory neurons to regulate axon termination (**Fig 5**). Consistent with these results, we found that CDK-5 colocalizes with the RPM-1 LD substrate 'trap' in the soma of ALM mechanosensory neurons (**Fig 6**). Thus, our results suggest that RPM-1 might inhibit CDK-5 in the soma to regulate axon termination at a distal site (**Fig 8**). Our findings are not the first example of RPM-1 affecting signaling in the cell body. RPM-1 was previously shown to interact with a nuclear anchoring protein, ANC-1/Nesprin, to affect axon termination [69]. Thus, our findings coalesce into a clear picture about how the RPM-1/FSN-1 ubiquitin ligase complex inhibits CDK-5 to influence axon termination.

### Cdk5, the RPM-1/FSN-1 ubiquitin ligase complex and disease

Our finding that CDK-5 is inhibited by the RPM-1/FSN-1 ubiquitin ligase complex prompts several considerations regarding disease. Importantly, Cdk5 and RPM-1 share notable links to pathological conditions and disease-relevant cellular processes. Both Cdk5 and RPM-1 are involved in axon degeneration [11,55,70–72], and genetically interact with players involved in Alzheimer's disease, such as Tau and tubulin acetyltransferase [42,49,73]. RPM-1 and Cdk5 regulate autophagy, a process broadly implicated in neurodegeneration [33,74]. Finally, Cdk5, RPM-1 and FSN-1 are all implicated in non-neuronal cancers [4,75–77]. Thus, our discovery that the RPM-1/FSN-1 ubiquitin ligase complex restricts Cdk5 to control axon development has unveiled several potential connections between these molecules and disease.

## Methods

### Genetics and strains

*C. elegans* N2 isolate was used for all experiments. Animals were maintained using standard procedures. The following mutant alleles were used: *rpm-1(ju44)* V, *cdk-5(ok626) III, fsn-1 (gk429) III, cdka-1(tm648) III* and *pct-1(tm2175) IV*. The following integrated transgenes were used: *muIs32* [P*mec-7*::GFP] II, *wdIs4* [P*unc-4*::GFP] II, *jsIs973* [P*mec-7*::mRFP] III and *jsIs821* [P*mec-7*::GFP::RAB-3] X. The following CRISPR alleles were used: *bgg6* [GFP::RPM-1 CRISPR], *bgg6 bgg40* [GFP::RPM-1 LD *C3535A, H3537A, H3540A* CRISPR], *bgg74* [*rpm-1* LD *C3535A, H3537A, H3540A* CRISPR], *bgg47* [GFP::FSN-1 CRISPR], *bgg52* [CDK-5::3xFLAG CRISPR], *bgg71 bgg77* [*cdk-5* KD *K33T, D144N* CRISPR], *bgg57* [CDK-5::mScarlet CRISPR]. Full details for all transgenic and CRISPR strains are shown in S1 Table. All mutants and CRISPR engineered strains were outcrossed to wild-type animals four or more times prior to experiments. Animals were grown at 23˚C for genetic analysis. Sequences of all genotyping primers for mutants and CRISPR strains can be found in S4 Table.

### Molecular biology and transgenics

The *cdk-5* genomic locus (pBG-354, 3kb promoter + open reading frame + 3'UTR) was amplified from N2 genomic DNA using iProof High-Fidelity DNA polymerase (BioRad), and

cloned into pCR-Blunt II-TOPO Vector (Invitrogen). *cdk-5* and *cdka-1* cDNAs were amplified from an N2 cDNA library made using random hexamers and SuperScript IV First-Strand Synthesis System (Invitrogen). Both cDNAs were subcloned into pCR8 and recombined using gateway technology (LR clonase II, Invitrogen) to generate final plasmids used for microinjection. Kinase-dead point mutations were generated in *cdk-5* cDNA using site-directed mutagenesis (QuikChange II XL Mutagenesis Kit, Agilent Technologies). All constructs were fully sequenced. Primer sequences used for all cloning can be found in S4 Table.

Transgenic extrachromosomal arrays were generated using standard microinjection procedures for *C. elegans*. Injection conditions and transgene construction details are specified in S2 Table.

## CRISPR/Cas9 editing and engineering

CRISPR editing individual point mutations and CRISPR engineering GFP or 3xFLAG tags into endogenous gene loci was done using *dpy-10* co-CRISPR and direct injection of Cas9 ribonucleoprotein complexes. Injection mixes contained tracrRNA (Dharmacon), crRNA (Dharmacon), repair template (PCR or ssODN (Extremers, Eurofins)), and recombinant Cas9 protein purified from Rosetta 2 *E. coli* (Millipore EMD, #71397). Ribonucleoprotein complexes were heated at 37°C for 15 min prior to injection. All gene edits were confirmed by sequencing and subsequently outcrossed at least four times to N2 animals. All injection conditions for CRISPR gene editing/engineering are shown in S2 Table. All crRNA and repair template sequences are shown in S3 Table.

## Proteomics

Full details for *C. elegans* proteomics was described previously [33], but we briefly revisit our procedures here. Mixed-stage animals transgenically expressing protein G—streptavidin binding peptide (GS)::RPM-1, GS::RPM-1 LD substrate 'trap' or GS::GFP (negative control) were harvested, frozen in liquid N2, ground to submicron particles using a cryomill (Retsch), and lysed under varying detergent conditions. Whole animal lysates were clarified by centrifugation to remove particulate matter, and incubated with IgG coupled Dynabeads (80mg of total protein with 500uL of beads) for 4h at 4°C. Sample quality control was performed by silver staining 10% of purified material. 90% of each affinity purified sample was briefly run on Tris-Glycine SDS-PAGE, stained with Coomassie, and whole samples were subjected to in-gel trypsin digestion. After acidification and desalting, peptide pools were dried and resuspended in 100 μl of 0.1% formic acid. 13 μl of each sample were loaded into an Orbitrap Fusion Tribrid Mass Spectrometer (ThermoFisher Scientific) coupled to an EASY-nLC 1000 system and on-line eluted on an analytical RP column (0.075 × 250 mm Acclaim PepMap RLSC nano Viper, ThermoFisher Scientific). Scaffold (Proteome Software) was used to validate MS/MS-based peptides and protein. All control and test samples were sent for mass spectrometry together. A total of 7 independent affinity purification proteomic experiments were performed for each genotype.

Candidate RPM-1-binding proteins were identified by affinity purification proteomics based on 3 criteria: 1) Candidate was detected in a minimum of two of the seven total proteomic experiments performed. 2) Candidate had 2x or more total spectra enrichment in the GS::RPM-1 or GS::RPM-1 LD sample compared to the negative control (GS::GFP). 3) Ribosomal and vitellogenin proteins were removed because they are widely regarded as proteomic contaminants.

## CRISPR-based native biochemistry

Mixed-stage animals for CRISPR engineered strains were grown in liquid culture (S complete media and HB101 *E. coli*) for 3 days at room temperature. Worms were harvested by low-

speed centrifugation, and bacteria/debris removed by 30% sucrose flotation centrifugation. Animals were subsequently washed three times with 0.1 M NaCl solution and clean, packed animals were frozen as pellets in liquid nitrogen. Frozen worm pellets (2.5-5g) were ground to submicron particles under liquid nitrogen cooling using a cryomill (Retsch) in the presence of EDTA-free protease inhibitor tablets (Roche). Grindates were lysed in four times volume lysis buffer (50 mM Tris HCl (pH 7.5), 150 mM NaCl, 1.5 mM MgCl$_2$, 10% glycerol, 0.1% NP-40, 1 mM DTT, 1 mM PMSF, 1x Halt Protease Inhibitor Cocktail (Thermo Scientific)). Whole animal lysates were rotated for 5 min at 4°C and high speed centrifuged (20,000×g) for 15 minutes to remove debris. Whole animal lysates (10–20 mg total protein) were incubated with 3–6 μg anti-FLAG (M2 mouse monoclonal, Sigma) or 0.6–1.2 μg anti-GFP (3E6, Invitrogen) antibodies for 30 min. Protein-antibody complexes were precipitated with 10–15 μL of protein-G agarose (Roche) for 4h at 4°C, and washed 3 times in lysis buffer. Samples were boiled in LDS sample buffer (NuPAGE, Invitrogen) and run on 3–8% Tris-Acetate gels (RPM-1) or 4–12% Bis-Tris gels (FSN-1 and CDK-5) (NuPAGE, Invitrogen). Gels were wet transferred onto PVDF membranes (Millipore) overnight (3–8% gels) or for 1h (4–12% gels) at 4°C. Immunoblotting was performed with anti-FLAG (1:1000 dilution, rabbit polyclonal, Cell Signaling) or anti-GFP (1:1000 dilution mouse monoclonal, Roche) antibodies. Immunoblots were visualized using HRP-conjugated secondary antibodies (1:20,000 dilution; GE Healthcare Life Sciences, Fisher Scientific) and ECL (1:3 diluted in TBS or undiluted Supersignal West Femto, Thermo Scientific). Blots were imaged with X-ray film, or KwikQuant or BioAzure CS600 digital imagers.

### Axon termination analysis

Axon termination in ALM neurons was evaluated using the transgene *muIs32* (P$_{mec-7}$GFP). ALM axons that showed short or long hooks extending beyond the normal termination point were considered failed axon termination defects. Axon termination in SAB neurons was evaluated using the transgene *wdIs4* (P$_{unc-4}$GFP). Failed axon termination in SAB neurons was defined as axons that formed a hook and were overextended at the tip of the axon. To quantify axon termination, young adult animals were anesthetized (10 μM levamisole in M9 buffer) on a 2% agar pad on glass slides. Coverslips were applied, and slides were mounted and visualized with a Leica DM5000 B (CTR5000) epifluorescent microscope (40x oil-immersion objective). For image acquisition, young adult animals were anesthetized (10 μM levamisole or 3% (v/v) 1-phenoxy-2-propanol in M9 buffer) using a 3% agarose pad on glass slides with coverslips. Images were acquired using a Leica SP8 confocal microscope or Zeiss LSM 710 (40x objective).

### CDK-5 and RPM-1 colocalization studies

For analysis of CDK-5::mScarlet CRISPR localization in ALM neurons, we used the transgene *muIs32* (P$_{mec-7}$GFP) to visualize cellular compartments including the soma, axon and terminated axon tip. For colocalization studies with RPM-1 LD, GFP::RPM-1 LD was injected into CDK-5::mScarlet CRISPR animals. GFP::RPM-1 LD and CDK-5::mScarlet CRISPR were simultaneously visualized using laser scanning confocal microscopy. For microscopy, young adult animals were anesthetized (10 μM levamisole or 3% (v/v) 1-phenoxy-2-propanol in M9 buffer) on 3% agarose pad on a glass slide and mounted with a coverslip. Images were acquired using a Zeiss LSM 710 confocal microscope (40x objective).

### Evaluation of CDK-5::mScarlet levels

For image acquisition, young adult animals carrying both CDK-5::mScarlet CRISPR and the transgene *muIs32* (P$_{mec-7}$GFP) were anesthetized (3% (v/v) 1-phenoxy-2-propanol in M9

buffer) on 3% agarose pad on a glass slide and mounted with a coverslip. Laser scanning confocal microscopy was used to collect images of ALM somas for wild-type animals and *rpm-1* mutants using a Zeiss LSM 710 (40x objective). For quantitative studies, Z projections of ALM somas for a given genotype were generated using Fiji/ImageJ. A region-of-interest (ROI) was drawn around the cell soma and ROI mean pixel intensity was calculated using ImageJ.

### ALM synapse analysis

ALM synapses were evaluated using the transgenes *jsIs821* (P$_{mec-7}$GFP::RAB-3) and *jsIs973* (P$_{mec-7}$mRFP) in adult animals grown at 23˚C. L4 larvae were grown overnight (16 hours) on fresh plates containing OP50 *E. coli*. For microscopy, these synchronized, young adults were anesthetized (3% (v/v) 1-phenoxy-2-propanol in M9 buffer) using a 3% agarose pad on glass slides and coverslips were applied. Slides were imaged (40x objective) using a Zeiss LSM 710 laser scanning confocal microscope. Z projections of single ALM branches were generated using ImageJ. Images of all genotypes were blinded for analysis using custom software. ImageJ fluorescence intensity plots of GFP::RAB-3 were created along the ALM branch. Fluorescence peaks with a value above 500 pixels were considered puncta for analysis purposes.

### Statistical analysis

**Axon termination analysis.** For analysis of axon termination in ALM and SAB neurons, statistical comparisons were done using a Student's *t*-test with Bonferroni correction on GraphPad Prism software. Error bars are SEM. Significance was defined as $p < 0.05$. Bar graphs represent averages from 4 to 10 counts (25–35 animals/count) obtained from four or more independent experiments for each genotype. Dots shown in plots represent average for individual counts.

**Synapse analysis.** For analysis of GFP::RAB-3 puncta in ALM neurons, data was subjected to an outlier test using the ROUT method (Q = 10%) and subsequent statistical comparisons were done using a Student's *t*-test with Bonferroni correction on GraphPad Prism software. Error bars are SEM. Significance was defined as $p < 0.05$. Bar graphs represent averages (4 to 11 animals) for each genotype. Dots shown in plots represent outcome for an individual animal.

**Quantitation of immunoblots.** For analysis of CDK-5 coIP with FSN-1 or RPM-1 LD, densitometry was performed using ImageJ software. Mean band intensity was determined for 5–6 replicates run simultaneously across 2 or more independent experiments. Each replicate is shown as an individual dot in graph. Statistical comparison was performed using a Student's *t*-test after evaluating data for normalcy using GraphPad Prism software. Wild-type values were normalized to 1 to simplify comparison of test samples relative to controls.

**Quantitation of CDK-5 levels in ALM neurons.** For analysis of CDK-5::mScarlet levels in ALM soma, statistical comparisons were done using a Student's *t*-test on GraphPad Prism software. Significance was defined as $p < 0.05$. Line represents averages (15 to 18 animals) for each genotype. Dots shown in plots represent outcome for an individual animal.

## Supporting information

**S1 Fig. A)** LC MS-MS spectra for two CDK-5 peptides identified in GS::RPM-1 LD samples. **B)** Highlighted in CDK-5 sequence are four unique CDK-5 peptides (red) identified in GS:: RPM-1 LD substrate 'trap' samples. **C)** LC MS-MS spectrum of one CDKA-1 peptide identified in GS::RPM-1 LD samples. **D)** Highlighted in CDKA-1 sequence are two unique peptides (red) identified in GS::RPM-1 LD samples.
(TIF)

**S2 Fig.** Quantitation indicates ALM axon termination defects are reduced in *cdka-1; rpm-1* double mutants compared to *rpm-1* single mutants. Means (bars) are shown for 4 or more counts (black dots, 20 or more worms/count) for each genotype. Error bars indicate SEM. Significance assessed using Student's *t*-test. * p<0.05.
(TIF)

**S3 Fig.** Quantitation of axon termination defects in ALM neurons for indicated genotypes. Reduced frequency of axon termination defects occurs in *cdk-5; rpm-1* double mutants, but not *pct-1; rpm-1* double mutants. Suppression of termination defects is not increased in *cdk-5; pct-1; rpm-1* triple mutants compared to *cdk-5; rpm-1* double mutants. Results suggest that CDK-5 and PCT-1 do not function redundantly during axon termination. Means (bars) are shown for 6 or more counts (black dots, 20 or more worms/count) for each genotype. Error bars indicate SEM. Significance assessed using Student's *t*-test. * p<0.05, ** p<0.01.
(TIF)

**S4 Fig.** **A**) Schematic shows region of ALM mechanosensory neurons imaged (red box). **B**) Representative images of ALM soma from wildtype and *rpm-1 (lf)* mutants. Shown is transgenic GFP expressed in mechanosensory neurons (P*mec*GFP; left, green), CDK-5::mScarlet CRISPR (middle, magenta) and merged image (right). **C**) Quantitation shows CDK-5::mScarlet levels in ALM soma are not different between wildtype animals and *rpm-1 (lf)* mutants. Means (grey lines) are shown for each genotype with individual data points (red dots) representing single animals. Significance assessed using Student's *t*-test. ns, non-significant.
(TIF)

**S5 Fig.** RPM-1 and CDK-5 function in parallel to regulate synapse formation in ALM mechanosensory neurons. **A**) Schematic shows primary axon and collateral synaptic branch of ALM mechanosensory neurons. Highlighted in green on collateral branch are presynaptic sites labeled by RAB-3::GFP. **B**) Shown are representative confocal images of presynaptic RAB-3::GFP puncta for indicated genotypes. For wildtype, images are shown for GFP::RAB-3 (upper left, green), RFP expressed in mechanosensory neurons (P*mec*RFP; upper middle, magenta) and merged image (upper right). **C**) Quantitation of GFP::RAB-3 puncta in ALM neurons for indicated genotypes. Note *rpm-1* mutants have reduced numbers of RAB-3 puncta, and these defects are enhanced in *cdk-5; rpm-1* double mutants. Means (bars) are shown for each genotype (black dots indicate individual worms). Error bars indicate SEM. Significance assessed using Student's *t*-test with Bonferroni correction. ns, non-significant; * p<0.05. Scale bar 10 μm.
(TIF)

**S1 Table. Transgenic and CRISPR Strains.**
(DOCX)

**S2 Table. Injection Conditions.**
(DOCX)

**S3 Table. CRISPR Targeting Sequences and Repair Templates.**
(DOCX)

**S4 Table. Primers used for genotyping and cloning.**
(DOCX)

## Acknowledgments

We would like to thank Dr. George Tsaprailis from the Proteomics Core of The Scripps Research Institute—Florida for his expertise and input. We thank the *C. elegans* Genetics Center for strains and the Wormbase genetic resource database.

## Author Contributions

**Conceptualization:** Muriel Desbois, Brock Grill.

**Data curation:** Muriel Desbois, Karla J. Opperman, Jonathan Amezquita.

**Formal analysis:** Muriel Desbois, Jonathan Amezquita.

**Funding acquisition:** Brock Grill.

**Investigation:** Muriel Desbois, Karla J. Opperman, Jonathan Amezquita, Gabriel Gaglio, Oliver Crawley.

**Methodology:** Muriel Desbois, Karla J. Opperman.

**Project administration:** Brock Grill.

**Supervision:** Brock Grill.

**Validation:** Muriel Desbois.

**Writing – original draft:** Muriel Desbois, Brock Grill.

**Writing – review & editing:** Muriel Desbois, Brock Grill.

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
