## [Decision Letter · Decision Letter 0]

11 Oct 2021

Dear Dr Grill,

Thank you very much for submitting your Research Article entitled 'Ubiquitin ligase activity inhibits Cdk5' to PLOS Genetics.

The manuscript was fully evaluated at the editorial level and by independent peer reviewers. The reviewers appreciated the attention to an important problem, but raised some substantial concerns about the current manuscript. Based on the reviews, we will not be able to accept this version of the manuscript, but we would be willing to review a much-revised version. We cannot, of course, promise publication at that time.

The Editors have read the manuscript and agree with the Reviewers' concerns. In particular, it will be important to address the following main points raised by one of more of the Reviewers:

Are CDK-5 protein levels increased in rpm-1 and fsn-1 mutants? Is CDK-5 ubiquitinated and is this blocked in rpm-1 mutants? Showing a direct effect of rpm-1 on CDK-5 ubiquitination and protein levels will strengthen the idea that CDK-5 is a target of the RPM-1 ubiquitin ligase and strengthen the partial genetic suppression data described by Reviewer #1.Clarify the interpretation of the data showing partial suppression of rpm-1 axon termination defects by cdk-5 and cdka-1, and the smaller effect of CDK-5 and CDKA-1 overexpression on axon termination versus rpm-1 mutants.Address whether CDK-5 also act downstream of rpm-1 to regulate its effects on synapse formation.

If you decide to revise the manuscript for further consideration at PLOS Genetics, please aim to resubmit within the next 60 days, unless it will take extra time to address the concerns of the reviewers, in which case we would appreciate an expected resubmission date by email to plosgenetics@plos.org.

[LINK]

We are sorry that we cannot be more positive about your manuscript at this stage. Please do not hesitate to contact us if you have any concerns or questions.

Yours sincerely,

Peter Juo

Guest Editor

PLOS Genetics

Gregory P. Copenhaver

Editor-in-Chief

PLOS Genetics

Reviewer's Responses to Questions

**Comments to the Authors:**

Reviewer #1: In this manuscript the authors' analyse their previous proteomics data and bring out the interaction between ligase dead RPM-1 and CDK-5. They confirm this interaction using in vivo biochemical studies using CRISPR-Cas9 strains. The authors’ further show a genetic interaction between cdk-5 and rpm-1 where cdk-5 mutants partially suppress the axon guidance defects in rpm-1 mutants in ALM and SAB neuron. The rescue experiments with ALM show that this is likely a cell autonomous rescue. The manuscript is well written and presented. However, I have multiple questions that relate specifically with the interpretation of the genetic experiments by the authors and the in vivo relevance of the interaction between CDK-5 and ligase dead RPM-1.

My comments are given below:

Major comments:

1. The authors show that CDK-5 interacts with Ligase Dead (LD) RPM-1, CDK-5 interacts with FSN-1 and the CDK-5, FSN-1 interaction is dependent on RPM-1 (LD). From the blots shown in figure 2 there is a clear indication that the RPM-1 (LD) version is what complexes with CDK-5, the blot also indicates that the WT RPM-1 does not interact with CDK-5. From his interaction it is unclear how CDK-5 is an RPM-1 substrate if the interaction is completely dependent on a ligase dead version of RPM-1. How does this translate in vivo and what is the relevance of this interaction in vivo. These points are not explained in the manuscript and are in my opinion very important points to address.

2. I feel that the authors are overstating the conclusions from their genetic experiments. In the first set of genetic experiments performed using the length of the ALM axon, that authors show that WT and cdk-5 mutants show no phenotype, about 75% of rpm-1 mutants show a failed termination or overshooting of the axon phenotype and around 50% of the cdk-5; rpm-1 show the overshooting phenotype. This would imply partial suppression of the rpm-1 phenotype by cdk-5 mutants. I feel that concluding from this experiment that RPM-1 inhibits CDK-5 would be flawed. A similar problem is seen in case of fsn-1 mutants, where the experiments indicate that cdk-5 suppresses the fsn-1 mutant phenotype and does not show that FSN-1 inhibits CDK-5. Again, similar problems are seen with cdka-1; rpm-1 double mutants. In fact the suppression here seems to be fairly minimal, indicating only a partial suppression of the rpm-1 phenotype by cdka-1.

3. The rescue experiments in figure 4 are useful in indicating that the suppression of rpm-1 is because of CDK-5 in mechanosensory neurons. The overexpression experiment indicates a possibility of CDK-5 inhibiting RPM-1 as the authors have stated, however it can be noted here that the overexpression phenotype of both CDK-5 and CDKA-1 together is only about 20% failed termination which is much less then what is seen in rpm-1 mutants, this would require a milder interpretation of the results seen. Further, the experiments in the manuscript allude to a complex interaction between CDK-5 and RPM-1 and parsing out this interaction may give a lot more insight into how the two molecules function.

4. The experiments in figure 5 are interpreted by the authors as RPM-1 specifically restricting the kinase activity of CDK-5, I think this would require showing a difference in CDK kinase activity in WT vs rpm-1 with more cells having kinase activity in rpm-1 mutants. I do agree that the experiments point to the fact that CDK-5 kinase activity is seen in ALM neurons for normal axon growth and termination.

5. The SAB neuron experiment confirms the fact that cdk-5 partially suppresses the axon termination defects seen in rpm-1. This is a nice addition to the ALM experiments but again does not conclusively show that RPM-1 inhibits CDK-5 in this system.

Minor comments:

1. The authors should indicate a list of primers in the supplemental tales detailing the primers used for making constructs and genotyping the strains used in this manuscript.

2. The plots indicating the proteomic data sets need to be better explained in the text or the figure legends. Plots B, D and D of figure 1 are difficult to understand and many terms like B ions and m/z are not described in the figure legends. Further, there is no methods section available for the proteomics experiment, this should be added to the methods. The authors do cite the method from their previous paper but a small section in this manuscript would help the reader.

3. Please describe the control CRISPR strains used (page 6, line 3).

4. For the ALM axon guidance experiments please show both rpm-1 like and WT like phenotypes of cdk-5; rpm-1 animals. Currently, only the WT-like phenotype is indicated in panel 3A.

5. Similarly in panel 6A it may help the reader to show both phenotypes seen in SAB axons of cdk-5; rpm-1 C. elegans.

Reviewer #2: In this manuscript, Muriel Desbois et,. al uncover a surprising role of E3 ligase rpm-1 in regulating CDK-5. Both rpm-1 and cdk-1 have profound roles in neuronal development, and this study is the first to connect those two genes. Over all, the studies were well designed, and the proteomic and genetic data are convincing. I only have a few questions as following:

1) the only missing evidence is whether rpm-1 can directly degrade CDK-5 through UPS system. This can be tested by ubiquitin IP (or IP CDK-5 for ubiquitin ) in control and rpm-1(lf) animals. It will also help if the authors have data to show rpm-1(lf) increase CDK-5 protein levels in neurons.

2) as both rpm-1 and cdk-1 are involved in synapse formation. It will be interesting to test whether CDK-5 also functions downstream of rpm-1 in synapse formation. This should be straight forward by testing the suppression ability of cdk-5(lf) on rpm-1(lf) synapse phenotype.

Reviewer #3: In this manuscript, Desbois et al showed that CDK5 is preferentially pull down by the kinase-dead RPM-1 but not by the wild type RPM-1 in worms. Biochemical analysis showed that RPM-1 interacts with CDK5, probably through FSN-1. Genetic analysis demonstrated that cdk5 loss of function rescues rmp-1 mutant phenotype, while gain of function of CDK5 mimics rpm-1 loss of function. These genetic analyses provided the most compelling evidence supporting the author’s argument and also suggested that the kinase activity of CDK5 is key in the RPM-1-mediated restriction of CDK5 activity in neurons. This study provided a potential novel mechanism in restraining CDK5 activity through the ubiquitin-mediated pathway.

I do have the following concerns:

Major:

1) One important outcome of the RPM-1-mediated regulation of CDK5 is not addressed (or even discussed): A] does the RPM-1/FSN-1 ubiquitin ligase targets CDK5 via polyubiquitination, which in turn promotes the proteasome-mediated degradation of CDK5? The data that in RPM-1 LD background, the interaction between FSN-1 and CDK5 is much stronger than in the wild type background seem to suggest that. However, easy experiments directly asking whether CDK5 abundance is elevated in rpm-1 or fsn-1 loss of function should be performed to clarify the issue and provide more solid and direct evidence to support the authors’ conclusion. The CRISPR allele CDK5-FLAG could be used if an endogenous CDK5 antibody is not available. In addition, whether the CDK5 kinase activity is enhanced in rpm-1 or fsn-1 mutants could also be tested. B] Another less-likely possibility is that RPM-1/FSN-1 mediates mono-ubiquitination of CDK5, which in turn suppress CDK5 kinase activity. Again, biochemical analysis of CDK5 abundance and SDS-PAGE migration pattern (in conjunction with Ubiquitin western blots) would provide a cleaner picture.

2) In addition to the biochemical IP experiments, imaging analysis of co-localization of FLAG-CDK5 with GFP-RPM-1 (WT or DL) or GFP-FSN-1 in SAB motorneurons would provide more evidence supporting the physical interaction.

3) A key point brought up by the authors is that FSN-1 bridges the interaction between RPM-1 and CDK5. It would be interesting to test whether RPM-1 LD could pull down CDK5 in the fsn-1 null mutant background.

Minor:

I would advise against using the term “in vivo, native biochemistry”. The analysis is completely in vitro when you use detergent to dissolve all the cellular and intracellular boundaries, in order to generate the cell lysate. The only difference from a typical CO-IP experiment is that, instead of using antibodies to IP the endogenous proteins, you generated endogenous epitope tags, which does not make the biochemical analysis in vivo.

**Have all data underlying the figures and results presented in the manuscript been provided?**

Reviewer #1: Yes

Reviewer #2: Yes

Reviewer #3: Yes

PLOS authors have the option to publish the peer review history of their article (what does this mean?). If published, this will include your full peer review and any attached files.

Reviewer #1: No

Reviewer #2: No

Reviewer #3: No

---

## [Decision Letter · Decision Letter 1]

17 Mar 2022

Dear Dr Grill,

We are pleased to inform you that your manuscript entitled "Ubiquitin ligase activity inhibits Cdk5 to control axon termination" has been editorially accepted for publication in PLOS Genetics. Congratulations!

Reviewers #1 and #2 were completely satisfied by your revisions. Reviewer #3 has one comment (see below). We recommend that you consider the comment and, if appropriate, add a supplemental figure as suggested - you can make this change as you prepare the final draft for the production team (the editorial team will not need to re-evaluate).

Yours sincerely,

Peter Juo

Guest Editor

PLOS Genetics

Gregory P. Copenhaver

Editor-in-Chief

PLOS Genetics

Comments from the reviewers (if applicable):

Reviewer's Responses to Questions

**Comments to the Authors:**

Reviewer #1: The authors have done multiple new experiments and changes to the existing text to answer all my comments. I commend the authors for the thorough job they have done. I recommend publication of this manuscript in its current form.

Reviewer #2: the authors have addressed all my questions. Congratulations!

Reviewer #3: 1) I appreciate the effort from the authors to address my first concern “whether CDK5 is directly targeted by the RPM-1/FSN-1 ubiquitin ligase complex”. The same concern is shared by another reviewer. Unfortunately, the results are not ideal. However, if the authors believe the abundance of CDK5 is not significantly changed in the rpm-1 null mutant, they should show this piece of data somewhere with the explanations in the discussion. I assume any potential reader will beg an answer to such an easily-testable question.

2) My other concerns are adequately addressed.

**Have all data underlying the figures and results presented in the manuscript been provided?**

Reviewer #1: Yes

Reviewer #2: Yes

Reviewer #3: Yes

PLOS authors have the option to publish the peer review history of their article (what does this mean?). If published, this will include your full peer review and any attached files.

Reviewer #1: No

Reviewer #2: No

Reviewer #3: No

**Data Deposition**

http://datadryad.org/submit?journalID=pgenetics&manu=PGENETICS-D-21-01223R1

**Press Queries**

---

## [Editor Report · Acceptance letter]

11 Apr 2022

PGENETICS-D-21-01223R1 

Ubiquitin ligase activity inhibits Cdk5 to control axon termination 

Dear Dr Grill, 

We are pleased to inform you that your manuscript entitled "Ubiquitin ligase activity inhibits Cdk5 to control axon termination" has been formally accepted for publication in PLOS Genetics! Your manuscript is now with our production department and you will be notified of the publication date in due course.

With kind regards,

Livia Horvath

PLOS Genetics

On behalf of:
